# Automatic detection of floating instream large wood in videos using deep learning

Janbert Aarnink[1], Tom Beucler[1,2], Marceline Vuaridel[1], and Virginia Ruiz-Villanueva[1,3]

[1]Université de Lausanne, Faculty of Geosciences and the Environment (FGSE), Institute of Earth Surface Dynamics (IDYST), Quartier UNIL-Mouline - Bâtiment Géopolis, 1015 Lausanne, Switzerland
[2]Université de Lausanne, Expertise Center for Climate Extremes, 1015 Lausanne, Switzerland
[3]University of Bern, Institute of Geography, Hallerstrasse 12, 3012, Bern, Switzerland

**Correspondence:** Janbert Aarnink (janbert.aarnink@unil.ch)

**Abstract.** Instream large wood (i.e., downed trees, branches, and roots larger than 1m in length and 10cm in diameter) has essential geomorphological and ecological functions supporting the health of river ecosystems. However, even though its transport during floods may pose a risk, it is rarely observed and poorly understood. This paper presents a novel approach to detect floating pieces of instream wood from videos. The approach uses a Convolutional Neural Network to detect wood automatically. We sampled data to represent different wood transport conditions, combining 20 datasets to yield thousands of instream wood images. We designed multiple scenarios using different data subsets with and without data augmentation. We analysed the contribution of each one to the effectiveness of the model using k-fold cross-validation. The mean average precision of the model varies between 35% and 93%, which is influenced by the quality of the data it detects. When using a 418-pixel input image resolution, the model detects wood with an overall mean average precision of 67%. Improvements of up to 23% mean average precision could be achieved in some instances, and increasing the input resolution raised the weighted mean average precision to 74%. We demonstrate that the detection performance on a specific dataset is not solely determined by the complexity of the network or the training data. Therefore, the findings of this paper can be used when designing a custom wood detection network. With the growing availability of flood-related videos featuring wood uploaded to the Internet, this methodology facilitates the quantification of wood transport across a wide variety of data sources.

## 1 Introduction

Instream large wood includes downed trees, root wads, trunks, and branches of at least 10 centimetres in diameter and 1 metre in length (Platts et al., 1987). It is typically recruited from forested areas within the river catchment by natural tree mortality, wind storms, snow avalanches, wildfires, landslides, debris flows, bank erosion as well as beaver activity (Benda and Sias, 2003). Stored wood within the river corridor plays a crucial role by trapping sediment, creating pools, and generating spatially varying flow patterns (Keller et al., 1995; Andreoli et al., 2007; Wohl et al., 2018). Therefore, instream wood is a crucial driver of the rivers' form and functioning and positively influences the diversity of the river ecosystem (Wohl et al., 2017). Although beneficial for biodiversity, wood can also be a hazard. During floods, large quantities of transported wood may accumulate at bridges or narrow river sections, blocking the channel and causing localised inundations (Lucía et al., 2015). Additionally, the

accumulation of wood can damage or eventually destroy bridges (Diehl, 1997; Lyn et al., 2003; De Cicco et al., 2018; Pucci et al., 2023). Costly wood removal efforts have long been the default mitigation strategy (Wohl, 2014), without considering the ecomorphological impact (Lassettre and Kondolf, 2012; Collins et al., 2012). However, these preventive efforts can even be counterproductive. For example, natural wood accumulations upstream from infrastructure can trap more wood transported during high-flow events, preventing it from accumulating at the critical infrastructure downstream (Ruiz-Villanueva et al., 2017). A more complex river system resulting from instream wood can also dissipate more flood energy than a channelised river (Curran and Wohl, 2003; Hassan et al., 2005). Human influence has impacted wood regimes and the river ecosystem by building infrastructure, channelling, and removing wood from rivers (Wohl et al., 2019). Therefore, it is crucial to understand instream large wood dynamics by assessing the quantity (i.e., wood supply and storage) and transport or fluxes. In addition to direct monitoring and crowdsourced videos of floods, recent advancements in understanding large wood dynamics have emerged through experimental studies in flumes and numerical studies (Panici, 2021; Innocenti et al., 2023). Estimating the quantity of wood in river systems and its temporal variation has gained traction over the last few years. However, in-field wood transport data remains scarce. As wood is mainly transported during floods, observations of transported instream wood are rare, and very few rivers are currently being monitored with that aim (Ghaffarian et al., 2020, 2021). Different techniques can help assess a river's wood regime in terms of transport, such as Radio Frequency Identification (RFID), high-resolution aerial surveys, and video monitoring (MacVicar et al., 2009). With RFID tags, individual pieces of wood are given a unique identity, and their movement can be registered and tracked. RFID tags can be used to quantify the percentage of wood that moves each year (Schenk et al., 2013). Attaching GPS loggers to pieces of instream wood is expensive and limited in temporal range, but it can give temporal data with a high frequency during high discharge events (Ravazzolo et al., 2015). Aerial data can detect stored wood and wood jams (Haschenburger and Rice, 2004; Lassettre et al., 2008; Sanhueza et al., 2018). However, the best methods to quantify wood transport are video-based because such methods provide high temporal and spatial resolution (Ghaffarian et al., 2020). Before introducing deep learning methods, conventional computer vision methods were used for object detection. These methods rely on feature extraction techniques such as edge detection, background subtraction, template matching or Histogram of Oriented Gradients (HOG) (Zou, 2019). Edge detectors use pixel-based filters to analyse changes in image intensity (Sun et al., 2022). They help detect the contours of an object. Background subtraction algorithms work well with static camera setups (Kalsotra and Arora, 2021). It models the background and subtracts the background model from the current frame. Template matching techniques involve overlaying a template image across the input image to find regions that match the template (Swaroop and Sharma, 2016). Much like edge detectors, the HOG extracts features by counting the occurrences of gradient orientation in certain portions of the image (Dalal and Triggs, 2005). Although robust, it requires careful tuning.

Using computer vision software combined with stationary cameras for detecting wood transport has provided a first insight into river wood dynamics (Lemaire et al., 2015; Zhang et al., 2021). The approach uses spatial and temporal pixel-level analyses to, for example, detect features like colours, edges, and moving objects. Its first feature is a mask to identify potential floating objects that differ in colour from the water's surface. Combining these features eventually allows to ascertain the presence of wood in the images. Even though the approach's utility was proven and is used to extract wood from videos at a few sites (Zhang

et al., 2021), it still requires manual, site-specific tuning. It is purposefully designed for a specific site to increase performance.

When creating a method for a specific location, it performs well on data with which it is designed but becomes too specific and complex to generalize over a wide variety of datasets. Furthermore, the current method requires the camera to be angled in a fixed position to extract the wood-detection features, which decreases flexibility. Even when tuned to a specific location, its performance depends on seasonal and weather conditions (Ghaffarian et al., 2021). Furthermore, measuring stations are limited by their spatial locations and rely on specific installation setups before a wood-moving event.

Developments in mobile technology have enabled millions of people to use high-quality video cameras. During extreme weather events, videos of floods are often posted online, which can be an exciting source for wood transport analyses. Citizen science projects like the Argentinian Storm Chasers project have demonstrated the use of home videos to analyse hydraulic conditions during a storm (Le Coz et al., 2016). Similarly, crowdsourced videos can analyse wood regime characteristics (Ruiz-Villanueva et al., 2019). However, quantifying wood transport from videos recorded from non-fixed standpoints presents

a challenge, as existing tracking methods fail to analyse crowdsourced footage due to their dependency on a stationary camera angle. Manual detection and quantification of large wood have been conducted in previous studies (Ruiz-Villanueva et al., 2022); however, this process is labour-intensive. Advances in machine learning methods can be applied flexibly and could allow widespread wood detection.

Convolutional Neural Networks (CNNs) have been identified as an effective method for object detection (Lecun et al., 2015),

showing success in remote sensing environmental monitoring applications (Li et al., 2020). These methods have been utilised for various tasks, including segmenting tree trunks in urban areas (Jodas et al., 2021), detecting floating plastic debris in rivers (van Lieshout et al., 2020; Àlex Solé Gómez et al., 2022), and monitoring river flow (Dibike and Solomatine, 2001). In the studies that highlighted the potential of deep learning for detecting and classifying objects in fluvial environments, transfer learning was used to classify static stored large wood in rivers from aerial imagery, achieving a recall rate of 93.75%, to

overcome data scarcity (Schwindt et al., 2024). Similarly, deep learning has proven effective in other aquatic contexts, such as fish species detection and weight estimation (Sokolova et al., 2023), further illustrating the adaptability of CNNs in riverine environments. However, the application of CNNs for automating floating wood detection remains under-explored, which we attribute to the lack of uniform training data (Maxwell et al., 2018; Shorten and Khoshgoftaar, 2019).

In this study, we propose using a You Only Look Once (YOLO) CNN as a flexible first approach to analyse floating wood

in videos from various sources, circumventing the limitations of current state-of-the-art wood detection algorithms, which are site-specific and require calibration. Our algorithm aims to detect and track floating wood pieces in any river under various conditions and from varying sources. Different video sources could include permanent monitoring cameras and handheld devices from witnesses of wood-laden flood events in rivers. This offers immediate applications such as computing wood fluxes to understand wood dynamics in rivers and practical uses like warning systems, flood hazard and risk assessments.

## 2 Methods

### 2.1 Convolutional neural network selection

Our Convolutional Neural Network comprises multiple convolutional layers that analyse video frames. Convolutions are used to extract hierarchical features from images to make predictions (Lecun et al., 2015). Features such as edges, corners, and textures are combined to determine the class of an object. The algorithm learns which features are necessary for classifying an object as instream wood. These detection features do not require individual hard coding but are developed by training the network with class examples. Depending on the architecture of the CNN, it can thus be several orders of magnitude more complex and theoretically more effective at detecting wood. Training a CNN demands substantial data, ideally from diverse sources under varying weather and flow conditions (Bengio et al., 2013).

The main Convolutional Neural Networks using deep learning include the Region-Based Convolutional Neural Network (R-CNN), the Single Shot Multibox Detector (SSD), CenterNet, and the You Only Look Once algorithm. The R-CNN introduced the concept of region proposal networks (RPN) that first extract a region of interest before classifying it (Ren et al., 2017). This two-step process generally results in longer processing times. The SSD method, in contrast, does not use the region proposal step but instead utilises multi-scale feature maps (Liu et al., 2016). Another approach that omits the proposal stage is CenterNet, which identifies an object as a pair of key points representing the centre and size of the object (Duan et al., 2019). The YOLO algorithm features a unified architecture and detects objects as a single regression task, from image pixels to bounding box coordinates and class probabilities. This enables YOLO to excel in speed and accuracy (Bochkovskiy et al., 2020). Its single-pass architecture allows it to consider the entire image context during the detection phase, which is advantageous for identifying multiple pieces of wood simultaneously (Redmon et al., 2016). Furthermore, the YOLO algorithm has undergone seven major updates over the past six years (Redmon et al., 2016; Wang et al., 2022). These frequent updates and improvements can benefit the development of a wood detection method. Therefore, for the main part of this study, we have chosen to train the fourth generation of the You Only Look Once network.

### 2.2 Data

Training a Convolutional Neural Network (CNN) that can detect wood in various conditions requires multiple steps. First, instream wood data is acquired and labelled. Subsequently, the dataset is trimmed and augmented to create a database of varying images containing instream wood. Once these steps are complete, a large part of the database is used to train the model, whilst a smaller part is used to validate the training performance. In this context, "database" refers to all the data used to create the CNN, whilst a "dataset" is a subset of the database consisting of all the data recorded by the same device at a specific location and date. Figure 1 gives an overview of the data collection and processing and demonstrates that we assess the performance of 14 different augmentation and sampling strategies from the datasets.

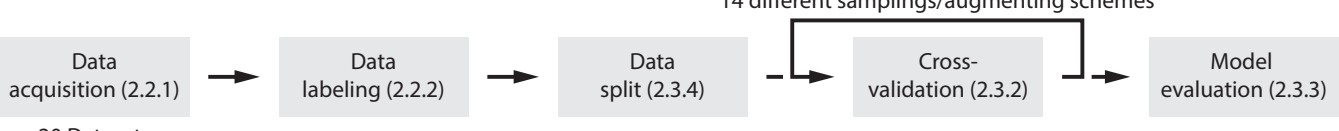

**Figure 1.** Overview of the methodology used for data collection and processing. Of the 20 labelled datasets obtained, six representative datasets are chosen for cross-validation (see Figure 4).

### 2.2.1 Data acquisition

For this study, we employed five low-cost cameras that were available to the authors, including three Android phones and two Raspberry Pi camera modules. The cameras are temporarily installed at various locations, on different days and times, with various orientations and resolutions (see table 1). The cameras were mounted to bridges and other stationary structures using makeshift supports to ensure a stable vantage point for capturing video footage of the floating wood. This method allowed for flexibility in positioning the cameras at various angles, depending on the bridge and the river section being monitored. We manually introduced wood upstream from the cameras into the river and let them record the wood passing by during a time window ranging from 30 to 90 minutes. We also used data from two locations in France that have been actively monitored by a permanent camera at the Allier since 2019 and the Ain Rivers since 2007 (Zhang et al., 2021; Hortobágyi et al., 2024). This data was used to analyse the wood flux and only contains natural instream wood occurrences. An operator-based visual floating wood detection method was employed to detect the wood. Labels were already available; however, as the labelling process involved only labelling the new pieces in each frame, the labels were insufficient for the method proposed in this paper. To test the model's performance after optimization, we gathered a test dataset. The test dataset consisted of 281 images with a resolution of $1280 \times 720$ pixels taken with a Xiaomi Mi 9 phone in timelapse mode. It was gathered at the river Inn at the moment of an experimental flood at its tributary, the Spöl River. The River Ecosystems Research Group at the University of Lausanne has actively studied this location for wood transport since 2018. It yields a valuable test dataset because an algorithm like ours could greatly reduce human labour. Lastly, carefully selected images of floating wood were added from online sources (purchased from istockphoto.com and dreamstime.com) to represent a small but diverse floating wood dataset. All images in this dataset are from different sources and in various locations. Additionally, instead of being recorded from a camera attached to a bridge, these images were captured by photographers. When this dataset is used as training data, it helps increase the variety of data sources. When used as validation data, it is a benchmark on how well the model generalizes wood. The final database consists of 15,228 images, each containing one or more pieces of wood. The images are separated into 20 different datasets, of which nine are shown in Figure 2.

### 2.2.2 Data labelling

Using both manual and automated approaches, labels are created on the data acquired to indicate where an instream piece of wood is located within each frame. Bounding boxes represent the four coordinates of a box's corners that fit around the piece

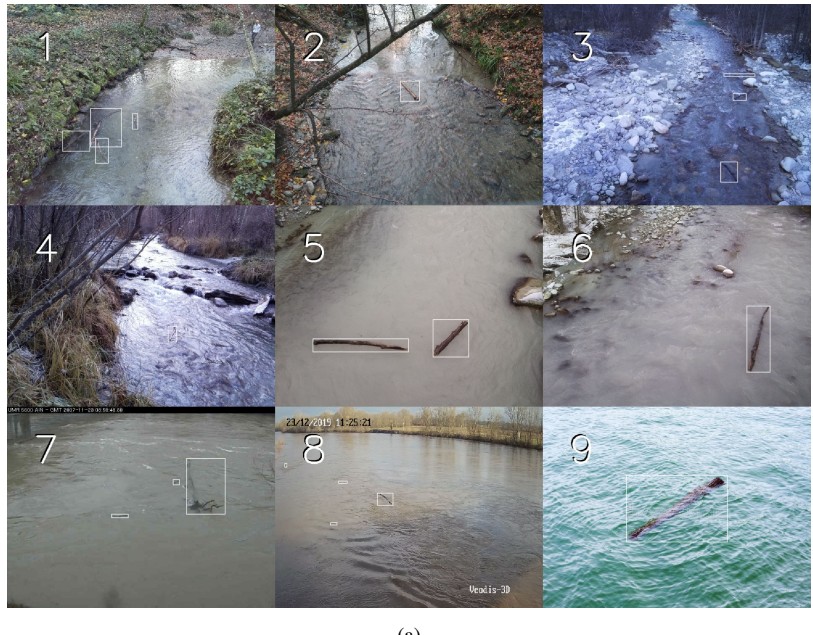

(a)

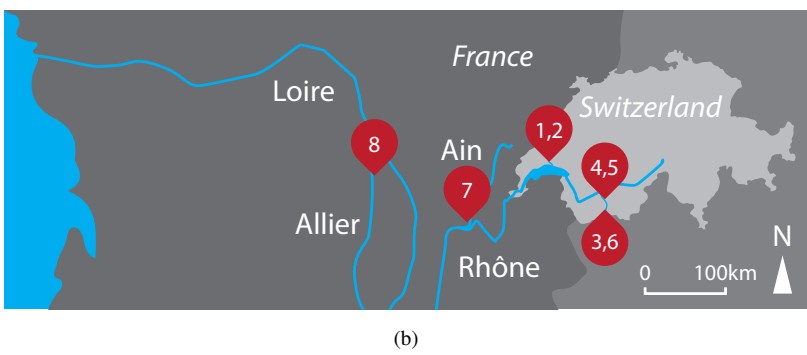

(b)

**Figure 2.** Examples from 9 of the 20 datasets. (a) Examples with bounding boxes around instream wood. (b) Locations of taken datasets. Images 1 (46.52373° N, 6.57729° E) & 2 (46.52296° N, 6.57577° E): La Chamberonne. Image 3 (46.04814° N, 7.48884° E): La Borgne d'Arolla. Image 4 (46.17966° N, 7.4187° E): Dixence. Images 5 (46.1612° N, 7.44079° E) & 6 (46.10975° N, 7.49428° E): La Borgne. Image 7: Ain River (image acquired by ENS Lyon). Image 8: Allier River (image acquired by ENS Lyon). Image 9: unknown location (purchased from iStock.com).

of wood (see Figure 2 for examples of bounding boxes). Initially, the labelling is manually performed using labelling software called LabelImg (Viso.ai, 2022). To expedite the process, we devised a pseudo-labelling method. Only 10% of the images in each dataset (1922 in total) were labelled by hand. A CNN (CenterNet, (Duan et al., 2019)) was trained to deliberately overfit that specific dataset by using only images from that particular dataset. With the CNN, the labels for the other 90% of images are created. Subsequently, we verified that all bounding boxes correctly indicated a piece of wood and adjusted the incorrect labels by going through all labels manually. It was verified that this method worked well in 11 out of the 15 cases in which we had

**Table 1.** Data acquisition statistics. For datasets 1-15, the devices were temporarily attached to bridges. Datasets 16-19 are taken from permanent monitoring stations. Dataset 20 consists of random samples found online.

| Dataset number | Amount of images | Amount of unique labels | Device | Resolution | Location | Number in Figure 2 |
|---|---|---|---|---|---|---|
| 1 | 1,429 | 3,743 | Raspberry Pi Camera | 1920x1440 | La Sorge, loc 1 | 1 |
| 2 | 601 | 1,598 | Raspberry Pi Camera | 1920x1440 | La Sorge, loc 1 | |
| 3* | 1,076 | 2,930 | Samsung Galaxy A4 | 3264x2448 | La Sorge, loc 1 | |
| 4 | 478 | 1,195 | Xiaomi Redmi 4X | 4160x3120 | La Sorge, loc 1 | |
| 5 | 344 | 674 | Xiaomi Redmi 4X | 4160x3120 | La Sorge, loc 2 | 2 |
| 6 | 2,478 | 5,436 | Raspberry Pi Camera | 1920x1440 | La Sorge, loc 2 | |
| 7* | 2,146 | 4,029 | Raspberry Pi Camera | 1920x1440 | La Sorge, loc 2 | |
| 8 | 191 | 343 | Samsung Galaxy A4 | 3264x2448 | La Sorge, loc 2 | |
| 9 | 18 | 28 | Xiaomi Redmi 2 | 3328x2496 | Borgne d'Arolla | |
| 10 | 138 | 256 | Raspberry Pi Camera | 1920x1440 | Borgne d'Arolla | |
| 11* | 1,046 | 2,116 | Raspberry Pi Camera | 1920x1440 | Borgne d'Arolla | 3 |
| 12 | 1,034 | 1,946 | Raspberry Pi Camera | 1920x1440 | Dixence | 4 |
| 13* | 157 | 180 | Raspberry Pi Camera | 1920x1440 | La Borgne | 5 |
| 14 | 2,340 | 4,472 | Raspberry Pi Camera | 1920x1440 | La Borgne | 6 |
| 15 | 1,236 | 2,232 | Samsung Galaxy A4 | 3264x2448 | La Borgne | |
| 16 | 116 | 229 | HDTV720P | 640x480 | Ain | 7 |
| 17 | 81 | 152 | HDTV720P | 640x480 | Ain | |
| 18* | 176 | 1,239 | Hikvision DS-2CD2T42WD-I8 | 1920x654 | Allier | |
| 19 | 134 | 353 | Hikvision DS-2CD2T42WD-I8 | 1920x1080 | Allier | 8 |
| 20* | 9 | 9 | Various | Various | Various | 9 |
| Average | 678 | 1,658 | | 2247x1673 | | |
| Total | 15,228 | 33,160 | | | | |

*Used as representative dataset.

abundant data and required minimal manual intervention. However, for the other four datasets, the CenterNet's performance was not sufficient to aid in the labelling of the other 90% of images, as the mean Average Precision (mAP, see section 2.3.3) was below 20%. Therefore, it would have required too much time and effort to manually create a completely labelled dataset; therefore, only the hand-labelled 10% of images were used.

### 2.2.3 Data analysis

After labelling, we obtained 15,228 fully labelled video screenshots with bounding boxes (see Figure 2) around 33,160 pieces of instream wood. When training a CNN, the goal is to have diverse data. Wood naturally floats and drifts with the current, or

becomes deposited and trapped by obstacles such as riverbanks, boulders, or trees. As a result, some videos record the same
piece of wood at the exact location for several minutes. Therefore, the data was trimmed. If, for subsequent frames, the labels
that encompassed the identified pieces of wood were almost identical (with the location, width and height of all bounding
boxes being within a certain, differing, percentage of each other based on visual assessment), only 1 of the frames was kept in
the database.

To prepare the data for analysis, from the database, all labels were cropped out of their corresponding images and resized
to greyscale images of a certain resolution. This resulted in a dataset of only the pieces of wood cropped from the images (for
an example see Figure 3). The pixel resolution was maximised to $80 \times 80$ pixels, the largest size in which all labels could be
processed with the available random access memory. This yielded images of all 33,160 pieces of wood in the database without
their surrounding image. Subsequently, the images were normalised and centred to eliminate circumstantial and camera-specific
white-balance differences. This means that the average pixel intensity of each picture was set to 128 and the maximum or
minimum pixel value to 255 or 0, respectively (see Figure 3). To analyse the variance in the data and perform clustering
in an unsupervised manner, we used a Python package called "clustimage" (Taskesen, 2021) along with custom scripts. The
analysis consisted of dimensionality reduction and clustering. First, a Principal Component Analysis (PCA) was applied to
reduce the dimensionality of the data. The high-dimensional data ($80 \times 800$ pixels) was transformed into a lower-dimensional
space while retaining 98% of the original variance. This reduced the complexity of the dataset, making it more manageable
for subsequent clustering. Following the PCA, we clustered the reduced dataset using a k-means algorithm. For four different
instances with 2, 4, 6 and 8 clusters, the silhouette score was calculated to evaluate the similarity within the clusters compared
to their dissimilarity across various clusters, effectively gauging the compactness and separation of the clusters.

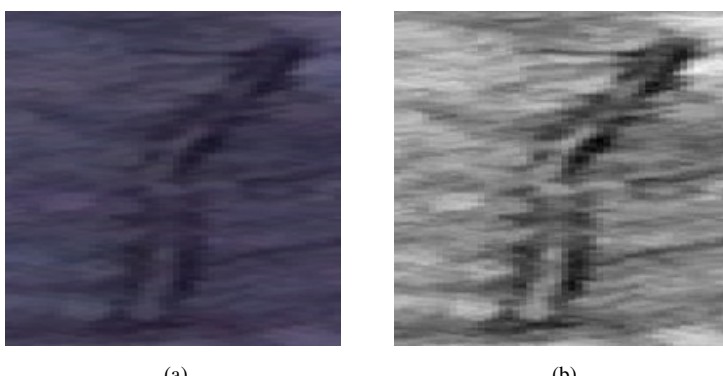

(a)                     (b)

**Figure 3.** (a) Original cutout and (b) greyscaled, normalised, and centred cutout (2020-11-29 Raspberry Pi 4 image 7411 label 2).

After the Principal Components Analysis, we performed a t-Distributed Stochastic Neighbor Embedding (t-SNE). The
stochastic nature of the t-SNE method means that although each run seemed to cluster similar samples, the exact output
graph differs each time. Therefore, we used it for visual interpretation purposes only. Furthermore, we compared the relative
size of the bounding boxes across datasets to understand the difference in the data.

### 2.2.4 Data split

Machine learning data is typically split into training, validation and test data. The training data trains the neural network to recognise patterns and make predictions. The validation data is used to tune the model's hyperparameters to ensure it performs well on out-of-sample data. The test data is unseen data used to benchmark the final performance of the CNN (Xu and Goodacre, 2018). Usually, all labelled data is combined, after which a certain portion, such as 90%, is randomly assigned as training data, whilst the other 10% is assigned as validation data. This ensures the training and validation data represent the overall data.

In our case, however, as the data comes from a limited amount of sources with common locations and camera angles, splitting the data using traditional methods might cause overfitting, resulting in overestimating the model's performance. A model that overfits performs well on training data but poorly on unseen data because it has learned the specifics of the training data too well. Multiple leave-one-out cross-validations can be used to mitigate overfitting, where one complete dataset is left out of the training data and used as validation data.

For feasibility purposes, six validation cycles were run. For each cycle, a single dataset was dropped for validation whilst the model was trained on the remaining 19 (see Figure 4). The six representative datasets were chosen to ensure diversity in location, camera angle, and time and reduce computational overhead by avoiding 20 validation cycles. Tests in section 2.3.1 demonstrate that sampling only 500 images from smaller than 500 images were chosen as representative datasets. For each training scenario, the performance was averaged over the six runs. This process was repeated for 14 different training scenarios as described in section 2.3.2.

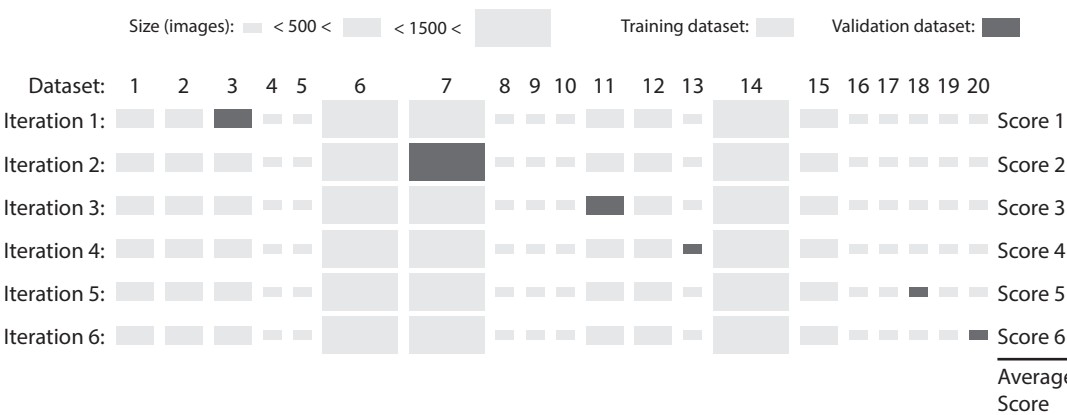

**Figure 4.** Cross validation scheme for one training scenario. This figure shows the distribution between the datasets used for training and those used for validation. The Y-axis contains all available datasets, and the X-axis contains the different training efforts. The dark grey dataset represents the validation data, while the other 19 were used to train the model. Eventually, the six scores are averaged into the validation score. The size of the rectangle represents the size of the dataset.

 ## 2.3   Machine learning

### 2.3.1   Training data size sensitivity

Before attempting to enhance performance across various scenarios, we assessed whether the entire dataset was necessary for model training. It was hypothesised that reducing the training data might not compromise model efficacy while potentially accelerating the training process. The 20 training datasets have an average size of 761 images and a median size of 411, with  some datasets containing almost 2,500 images. To reduce computational demands, we conducted tests to evaluate how the number of images sampled per dataset affects training performance. We conducted two tests with a fourfold difference in the total number of training images compared. In the first test, 2,000 images were augmented and sampled per dataset. Since the datasets did not all contain this exact number of images, they were randomly over or under-sampled to reach this figure. When oversampling, no additional data was introduced; instead, it ensured that dataset sizes were equalised, thereby preventing biases  in the rewards towards any particular dataset. We applied the same approach for the second test but with 500 images per dataset. Thus, a total training sample size of 38,000 images for the first test (2,000 images per dataset across 19 training datasets) was compared to 9,500 for the second test (500 images per 19 datasets). Following this comparison, we determined whether using less total data would decrease the model's performance.

### 2.3.2   Cross-validation procedure

 When training the YOLO CNN, the training and validation images are downscaled to a resolution of 416x416 as standard and consist of three RGB colour bands. A series of experiments were conducted to enhance the model's performance. Including the baseline, fourteen different testing scenarios were performed to test the model's sensitivity to stationary frames, dataset size, data augmentation, and data quality. As CNNs have not yet been trained for detecting instream large wood, thorough testing of different training strategies is crucial. To enhance model performance, the database size can be expanded by adding more  labelled images or through synthetic augmentation of existing data. Although more effective with image classification practices, augmenting data for object detection has been known to improve model performance by up to 2.7 percentage points for mean Average Precision (mAP) in some cases (Zoph et al., 2020). The augmentation practice, however, attracts less research attention because it is considered to transfer poorly to different datasets. Apart from augmentation, employing various sampling strategies may also improve the algorithm's detection performance. The data used for wood detection can also exhibit diverse camera  angles, pixel sizes, and proximity to the stream (see Table 1). To determine the most effective sampling and augmentation strategies for different data types, 14 models were trained and evaluated against a baseline model. The baseline model was trained using only the labelled images without any modifications. The other 13 scenarios are detailed below:

1. **Trimmed, testing sensitivity to handling stationary frames**: When labels were similar in at least three subsequent frames, the images were deleted from the database. Determining the exact pixels where a bounding box began and  ended was sometimes challenging (e.g. when part of a log was underwater and its end was not clearly visible). As a result, bounding boxes around an immobile object could vary from frame to frame. To account for this, detections were

considered similar when all bounding boxes were within 4% of their subsequent x and y locations in the frame and 30% of their width and height. These thresholds were determined manually by testing various percentages on cases where multiple stationary logs were detected in successive frames. In this scenario, 13,375 images from the total database were kept.

2. **Sampled V1, testing sensitivity to dataset size ("min500 max1200")**: As small datasets can be undersampled compared to large datasets, in this scenario, we sampled a minimum of 500 images per dataset and a maximum of 1,200 images per dataset. If the dataset was smaller than 500, we oversampled images randomly and added the duplicates to the dataset until we reached 500. We did not use all the data if the dataset was larger than 1,200. These numbers were chosen because when applying this sampling method to the 20 datasets, the total amount of labelled images was 15,257, similar to the total database (15,228 images).

3. **Sampled V2, testing sensitivity to dataset size ("750")**: In this sampling scenario, to sample equally from every dataset, 750 images from each were used. The total number of labelled images was 15,000, similar to the total database (15,228 images).

4. **Sampled V3, testing sensitivity to dataset size ("min500")**: To not delete data, in this scenario, only the small datasets were randomly oversampled to have a size of at least 500 images. As we kept all data in the other datasets, the total data size was larger than the baseline.

5. **Augmented V1, testing sensitivity to data augmentation ("mirrored rotated all")**: To increase the diversity of the data, the images were all used, and duplicates were mirrored and/or rotated. The rotation was kept between -15 and 15 degrees; in practice, the river almost always appeared at the frame's bottom. This was done because the data that must be analysed would normally also have the river on the bottom of the frame. In this scenario, the dataset contained twice the amount of images (30,456) as the baseline because a duplicate of each image was augmented randomly. Each image had a 50% chance to be mirrored. So approximately 7,614 images were mirrored, whilst all 15,228 copies were randomly rotated between -15 and 15 degrees.

6. **Augmented V2, testing sensitivity to data augmentation, ("mirrored rotated randomly")**: To increase the diversity of the data, the images were randomly selected to be mirrored and/or rotated. The rotation was kept between -15 and 15 degrees. In 50% of the cases, an image was mirrored (a total of approximately 7.614), and in 50% of the cases, the image was mirrored randomly between -15 and 15 degrees. The dataset for this scenario consisted of 15,228 images.

7. **Augmented V3, testing sensitivity to data augmentation ("only mirrored")**: The images were randomly mirrored in 50% of the cases to disentangle the mirroring and rotation effect. The dataset for this scenario consisted of 15,228 images.

8. **Augmented V4, testing sensitivity to data augmentation ("only rotated")**: The images were randomly rotated be-tween -15 and 15 degrees in 50% of the cases to disentangle the mirroring and rotation effect. The dataset for this scenario consisted of 15,228 images.

9. **Added V1, testing the sensitivity to data quality, added high definition non-floating wood**: In an attempt to increase the models' understanding of wood, we added photos of instream wood laying in the mostly dry riverbed to the database. A total of 167 photos containing at least one wood sample were added. The added data had pixel dimensions 4608 by 3456 and was higher quality than the other 20 datasets (see Table 1). Here, the influence of bounding box size and data quality was evaluated.

10. **Added V2, testing the sensitivity to data quality and diversity, added 12 datasets**: At a later stage in the testing process, from videos found online, a subset of the frames were labelled and added to the training database. A total of 10 datasets ranging from 8 to 118 images per dataset were added from locations in North America, New Zealand, and Switzerland. Additionally, two more self-gathered datasets from different sources containing 207 and 499 images were added. As taken from the Internet, the 1206 images added in this scenario were compressed and have an average pixel resolution of $1650 \times 1133$. Therefore, the added data quality was worse than the original 20 datasets. The added data is indicated with the letter 'A' in Figure 5. The A 11 and the A 12 descriptors are the self-gathered extra datasets.

11. **Removed, testing the sensitivity to data quality, removed worst performing datasets**: As lower quality data can weaken the models' understanding of wood, in this scenario, the quality of the data can be analysed by the effectiveness of the model trained in the base scenario to detect samples. The two datasets with the smallest relative bounding box size were assumed to include the fewest details and were of the lowest quality (see Figure 5). The three lowest-quality datasets were 12, 18, and 19. In this scenario, we removed datasets 12 and 19 from the training data to see whether the other datasets' detections improved. We considered dataset 12 a location where one would not likely monitor large wood. Also, as dataset 18 and 19 were taken from the same source and we wanted to keep data variability, we did not remove dataset 18.

12. **Merged, testing the sensitivity to adding a time component, merged three images into 1**: Because often the distinc-tion between a piece of instream wood or flow features like eddies and waves was not clear from a single image, in this scenario we merged three images into one image after converting them to greyscale. Therefore, instead of regular Red, Green, and Blue bands, the model was trained on greyscale images at T-1, T, and T+1, with T the timestep at which we were detecting. This was hypothesised to aid the detection as waves and eddies change during a short timestep, whilst the wood does not.

13. **Double Resolution, testing the sensitivity to increasing the input image size to the model, from 416 to 832 pixels**: A CNN was trained based on a specific pre-defined image resolution. As standard, from a higher resolution (between $640 \times 480$ and $4160 \times 3120$, see Table 1), images were resized to a $416 \times 416$ image resolution before they were used in training and validation. Decreasing the images could result in losing details, especially in cases where the relative size

of the wood pieces was small. Therefore, we evaluated the model's sensitivity to the input image size in this scenario by resizing the resolution to $832 \times 832$ pixels instead, retaining more details.

### 2.3.3 Model evaluation

Generally, a commonly used metric for object detection tasks to evaluate performance is mean Average Precision (mAP) (Tian et al., 2024), which combines three different measures: precision, recall, and intersection over union (IoU) (Zheng et al., 2020).

Recall is the percentage of wood pieces detected by the algorithm out of all the logs that pass by. Precision indicates whether the piece the CNN detected was indeed instream wood. The object detection algorithm outputs either no bounding boxes or (multiple) bounding boxes for each image. Each bounding box indicates the outer limits of the object and has a confidence percentage corresponding to how certain the model is in its detection. More bounding boxes are classified as a detection when lowering the confidence threshold. Hence, the recall increases, and the precision decreases. The changes in precision and

recall based on the threshold can be displayed in a precision-recall curve. The surface under the curve can be translated into a single average precision (AP) value for a specific IoU. However, this value does not compare different IoU thresholds. The IoU compares the label with detected bounding boxes by dividing their overlap by their combined total surface. For each IoU value, a different precision-recall curve can be created, resulting in different APs. When all different APs based on different thresholds and IoUs were combined into a single value, we got the mAP, which ranged between 0% and 100%. With an upper

limit of 100% mAP, the model would have labelled every instance of instream large wood exactly as the humans that labelled the training data. However, as human labelling is imperfect, the mean Average Precision was still not an objectively perfect performance index.

     Different applications of object detection call for different thresholds in recall, precision, and IoU, depending on the consequences. Depending on the large wood regime of a specific river, more emphasis can be placed on either recall or precision.

When the amount of wood passing is very low, e.g., one piece of wood a month, increasing recall can ensure that a piece is not missed. However, when using a too-high sensitivity, the model could wrongly detect wood in each frame, forcing the user to look at every image and delete all the false detections.

     We trained a model for all 14 training scenarios and validated it with the six validation datasets. The training was performed in epochs, which represented the number of times all the training images were used to train the model. The model was validated

using the validation data after a predetermined amount of epochs during the training process. The model's performance on the validation data was stored as mean Average Precision. During the same training session, the model can have several tests that perform similarly but also have an outlier in performance that cannot be reproduced. To account for this and not overestimate the performance, we used the best two validations mAPs to determine the performance of each training run. Furthermore, different training sessions using the same data can yield varying performance results, as the model may converge to different

local optima depending on the initialization and training dynamics. Therefore, we run each test three times to compare different training scenarios. Hence, we obtained an average of six best mAPs for each scenario.

     Additionally, on a small subset of training scenarios, a newer YoloV7 model was trained to compare the results of different models on the same data. A final model was trained after determining which training strategies worked best for which data

types. This model was tested on the test dataset described in section 2.2.4 that had never been used in any of the analysis and training efforts. In this way, the test dataset represented a case in which an unrelated wood monitoring study would use the model for out-of-the-box detection.

Finally, neural networks for object detection are often considered black boxes, which decreases their trustworthiness. To increase transparency, algorithms were developed to reverse the detection process and find the input pixels from the image weighted the highest when the process decided whether or not to detect an object. For the YoloV4 algorithm, we used a Python package called Yolime (Sejr et al., 2021) for this purpose. Yolime explains YOLOv4's object detection using LIME (Ribeiro et al., 2016). For each prediction, LIME perturbs the input data and observes prediction changes to highlight the image pixels that most influence the wood detection outcome. This process makes YOLOv4's predictions more understandable and, therefore, more trustworthy. Various instream wood samples from the database were hand-picked and Yolime was run to determine which pixels were most heavily used by the model to detect wood. The algorithm provided insights into which features of the image it identifies as characteristic of a piece of floating wood. Combined with data quality analysis, this understanding can help explain variations in model performance across different training scenarios.

## 3   Results

### 3.1   Training data: diverse but still clustered

The data used in this research appeared to be diverse. The PCA (Principal Component Analysis) yielded a low silhouette score and a visual inspection of the t-SNE (t-Distributed Stochastic Neighbor Embedding) plot revealed only small clusters of similar data (see Appendix A2), suggesting the presence of duplicates within the data. However, the analysis also uncovered similarities within each dataset. A comparison of the average bounding box sizes across datasets revealed distinct differences. Figure 5 illustrates the relative size of the bounding box compared to the overall image size, which was obtained by calculating the total amount of pixels in the envelope of the bounding box divided by the total amount of pixels in the image. The image highlights discrepancies in the sizes of labelled pieces of wood across different datasets. To adjust for the exponential distribution of calculated surface areas, we applied a square root transformation to the bounding box area for better visualization. The graph indicates that datasets 12, 18, and 19 of the original database had lower quality. This is the case because, in these cases, the size of the samples was small, and in the case of datasets 18 and 19, the camera was also located relatively far away. Examples of the difference in bounding box sizes between dataset 1 and dataset 12 are displayed in Figures C1 and C2 in the appendices.

### 3.2   Training results: database configuration matters most

First, analyses showed that the model's performance did not increase by oversampling data from 500 (9,500) to 2000 (38,000) images per dataset. The best mAP was similar for both tests. Figure 6 shows the difference between a training instance with 2,000 augmented images per dataset on the left and 500 augmented images per dataset on the right. It shows that increasing the data for a training instance slows down the model convergence without increasing eventual mAP.

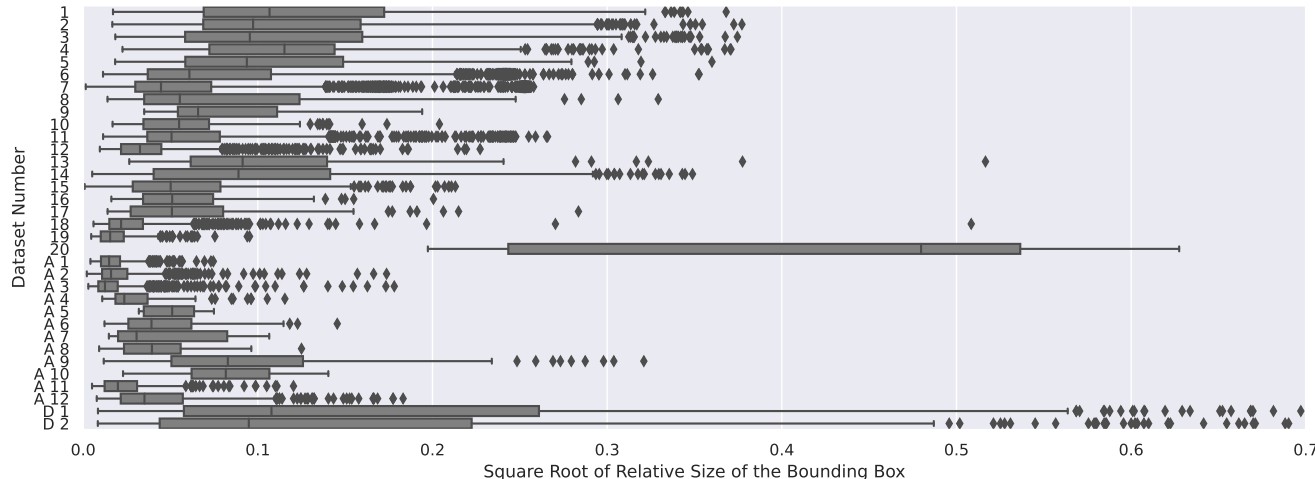

**Figure 5.** The relative size of the wood pieces compared to the image size per dataset. The relative size is represented by the square root of the surface of the bounding box sizes divided by the square root of the total image size. The square root of the relative size is shown to facilitate the interpretation of the figure. The datasets from table 1 are indicated by the number. The 'A' indicates datasets that are added in scenario 10. The 'D' indicates the dataset added in scenario 9.

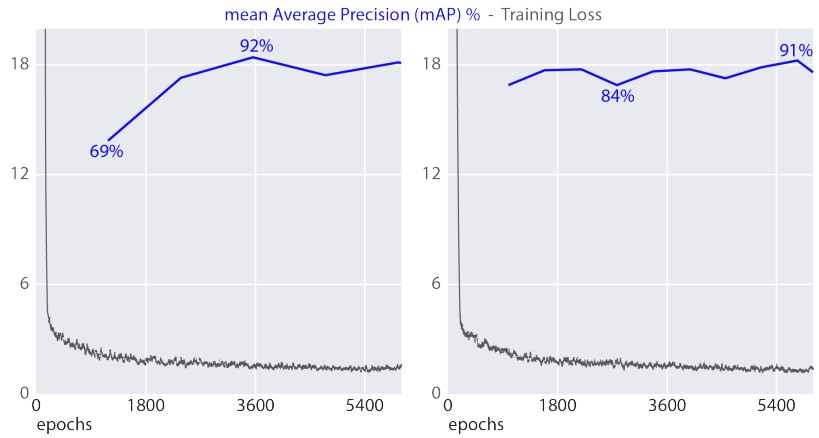

**Figure 6.** Training performance of dataset 13 (see Table 1) when using 2,000 images (left) and 500 images (right) per dataset (respectively 38,000 and 9,500 training images used). Blue: mean Average Precision, Grey: Complete Intersection-Over-Union Training Loss (Zheng et al., 2020)

Table 2 shows the training results of different scenarios. When using the total amount of 15,228 labelled images, the average performance in mAP was 63.42%. The following columns show the difference in performance (mAP) of the base scenario compared to the 13 test scenarios based on the six best performances. At the bottom of the table, the average of the six mAPs

**Table 2.** Neural Network performance for the base scenario in mean Average Precision (mAP, see section 2.3.3) and relative change in mAP percentage points when using the 13 different training scenarios. mAP changes of more than three percentage points, positive or negative, are indicated with a star.

| Scenario (dataset number) | Base Jpgs (mAP) | 1 Trimmed | 2 Sampled V1 | 3 Sampled V2 | 4 Sampled V3 | 5 Augmented V1 | 6 Augmented V2 | 7 Augmented V3 | 8 Augmented V4 | 9 Added V1 | 10 Added V2 | 11 Removed | 12 Merged | 13 Double Res |
|---|---|---|---|---|---|---|---|---|---|---|---|---|---|---|
| Sorge Samsung Galaxy A4 (3) | 80.33 | 0 | -3.33* | -2.67 | -1.33 | -1.5 | -2.17 | -1.83 | -2.5 | -1.17 | -1.83 | -1.33 | -18.83* | 3 |
| Sorge Pi 4 (7) | 71.17 | -0.17 | -1.67 | -3.67* | -0.83 | -2.17 | -3.83* | 0.17 | -1.67 | -0.17 | -1.17 | -1 | -3.17* | 5.17* |
| Borgne d'Arolla Pi 4 (11) | 43.67 | 5* | 14.67* | 13.67* | 15.33* | 4.83* | 0 | 0.5 | -2.5 | -1.17 | -1 | 2.5 | 6.17* | 17.17* |
| Dixence Pi 4 (13) | 92.83 | 0.5 | -0.17 | -1.67 | 0.67 | 1.17 | 2.33 | -0.33 | 2.17 | 0.17 | -0.67 | 0.17 | 1.17 | -2.17 |
| Allier (18) | 35.33 | -0.5 | -3.83* | -3.67* | -3.67* | -3.5* | -1.17 | 0.33 | -1.83 | -0.33 | 3 | -19* | -1.5 | 14* |
| Random Images (20) | 57.17 | -6.83* | 2 | 3.33* | -4.67* | -11.33* | -9.83* | -1.17 | -16.5* | 22.83* | -13.83* | 11.33* | -** | -22.33* |
| Average | 63.42 | -0.33 | 1.28 | 0.89 | 0.92 | -2.08 | -2.45 | -0.39 | -3.81* | 3.36* | -2.58 | -1.22 | -** | 2.47 |
| Weighted Average | 66.7 | 0.96 | 1.40 | 0.29 | 2.42 | -0.50 | -2.38 | -.21 | -1.94 | -0.55 | -1.14 | -0.91 | -** | 7.27* |

** dataset 20 does not contain a time series, and it is therefore not possible to merge

and the weighted average are shown. As there was a large variety in the sizes of the datasets, to not overestimate the importance of small datasets, the weighted average compensated for the relative size of the datasets.

Dataset 20, despite its small size, exhibited large variability because the images were sourced from different locations. This variability makes this dataset particularly useful for assessing the model's ability to generalize the concept of wood and detect it across diverse conditions. On the other hand, the larger datasets include cameras mounted to bridges and will, therefore, be a better representation of the primary use of the algorithm. Therefore, the weighted average is a better performance metric for the practical use of the algorithm.

Table 2 also shows the increase in model performance when changing the sampling strategy. When adding data from videos of floods containing instream wood found on YouTube and Twitter (scenario 10), the average mAP for all datasets went down by 2.58 percentage points. This decrease in performance can arguably be attributed to the low quality of the data being confusing to the model. This observation was also strengthened by the reduction in performance of this scenario when validating the high-definition random wood images dataset. Additionally, the model performed better when tested on the Allier River dataset, which mainly contains smaller (lower quality) samples of wood (see Figure 5). Instead of adding lower quality data, we added high-definition data of non-floating wood to the training data (Scenario 9), the significant increase in performance when validating dataset 20 was explained by the algorithm generalizing the concept of wood. This increased the average performance but decreased the weighted average performance as the overall average label sizes of dataset 20 were relatively small.

### 3.3 Test results

After analysing the scenarios, a novel test dataset (see section 2.2.4) was introduced to evaluate the model's performance, recorded during a flood event in the River Inn (see Figure 7). Notably, the model had never used this dataset during its training phase, and no adjustments were made to hyperparameters based on this new data. The mean average precision (mAP) from the

flood dataset was 61%, when using the base scenario. Increasing the input resolution of the model to $832 \times 832$, as per scenario 13, did not increase the performance (60.5% mAP).

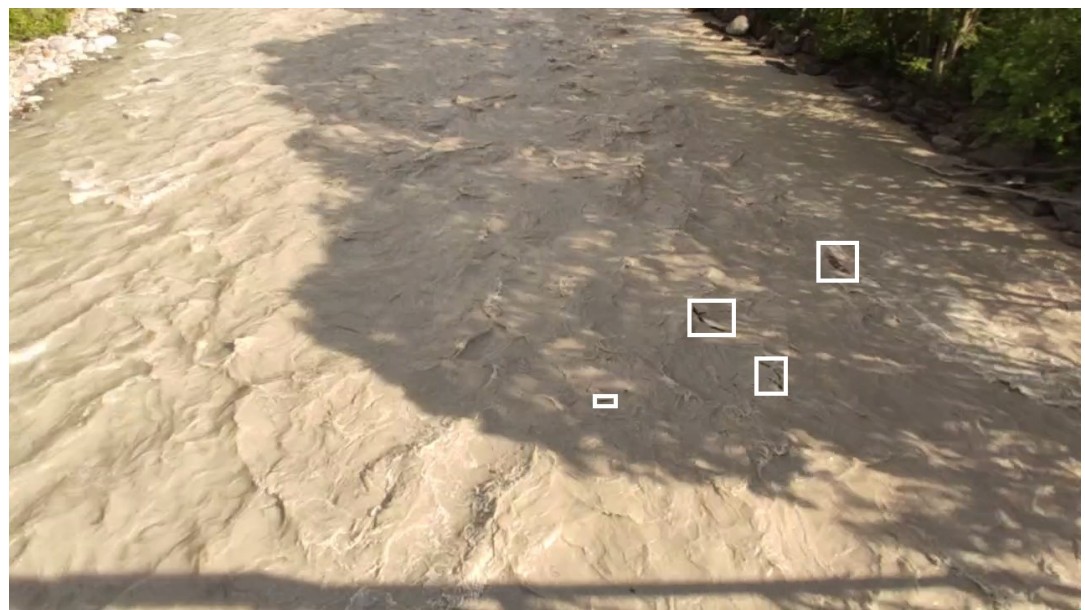

**Figure 7.** Example frame of the test dataset taken at the Inn River in June 2023. The white bounding boxes envelope pieces of instream wood. A camera attached to a bridge was recording in timelapse mode.

## 4 Discussion

### 4.1 Effect of data quality on performance and sampling strategies

Our research led to the development of a comprehensive database containing labelled videos of instream wood. The labels covered a broad spectrum of sizes, particularly when comparing different datasets. They ranged from clearly identifiable downed trees with distinct features such as bark, branches, and brown colour to bounding boxes resembling less defined shapes that spanned only a few pixels in both dimensions. Additionally, distinguishing water waves and eddies from pieces of instream wood can sometimes be challenging during labelling, as an image is a snapshot. At that moment, a wave ripple can resemble a piece of wood.

With a weighted average of 66.7% mAP, the model's performance was slightly better than the results of approximately 50% mean precision that van Lieshout et al. (2020) found when creating a plastic detection algorithm and testing it on an unseen new location. The table shows the results of the different training scenarios explained in section 2.3.2. The sensitivity analysis indicated that increasing the dataset size to 2,000 images per set did not enhance performance compared to 500 images per set. This suggests that oversampling from a limited number of data sources does not improve model performance, as the model

tends to overfit specific instances. Consequently, when training a custom model or making future enhancements, it is sufficient to utilize 500 labelled images. This threshold can optimise resource allocation and efficiency.

The proposed method does not require tuning of the algorithm for different sites, which allows its implementation on a range of different data sources. However, the training database, although extensive in size, is still limited in terms of the number of sources. The test where we see the best performance is during scenario 6 when using dataset 13. In this test, the model performed at a mean Average Precision (mAP) of 95.2%. The model's worst performance was on dataset 18 using scenario 11 at 16.3% mAP. From this large difference in performance, it can be argued that the data quality and diversity are still limiting

factors of the approach. This was reinforced by qualitative analysis of the quality of the datasets. As lower-quality data might confuse the model, scenario 11 was performed where the datasets with lower quality (datasets 12 and 19) were excluded from the training data. The results (table 2) show a weighted average decrease of 0.91 percentage points. This decrease was primarily linked to the worse performance of the model on the Allier River (dataset 18) where the decrease was large (19 percentage points) because the excluded dataset was taken from the same data source on a different day. Therefore, it can be argued that

the model was still shown to overfit the training data even with the precautionary measures. Reasoning the other way around, the -19 percentage points can be interpreted as +19 percentage points when adding data from the same scene on another day. This particular removed dataset contained 176 images. This shows a more practical implication for researchers. When starting a new monitoring project, it is good practice to label and add as few as 200 images to the larger database. In this way, one can train a site-specific wood detection algorithm which is shown to perform better than the model out of the box. The above

findings also showed that although the validation data used to calculate the mAP was taken from a different data source than the training data, there was still similarity. Data from the same camera on different days or the same location and date taken with different cameras were shown not to be completely different.

    Additionally, the results showed that the model's performance increased when using different sampling strategies (scenarios 2, 3 and 4). This was primarily because the detection on representative dataset 11 was more accurate. Dataset 10 was taken with

the same camera angle at a different time. The scene looked different, with a sunny background and sharp shadows, compared to the evenly illuminating overcast of dataset 11. However, oversampling the 138 labelled images from dataset 10 positively affected the models' performance on dataset 11 in all three scenarios. This underlines the need for careful sourcing of training datasets and shows the influence of dataset sizes. The model is trained to optimise performance on all training data and will, therefore, be biased towards the larger datasets. For the model to detect wood in a wide variety of scenes during the training

phase, it is helpful to have equal dataset sizes, whilst a custom model for detecting wood from a single camera angle can benefit from oversampling from that specific scene. Furthermore, the data trimming from scenario 1 seemed to have limited effect.

    In scenarios 5, 6, and 7, data augmentation was applied to introduce variability in the form of mirroring and slight rotations. Scenario 5 mirrored and rotated all images with slight rotations ranging from -15 to 15 degrees, while in scenario 6 images were rotated and mirrored randomly. In scenario 7, 50% of the images were randomly mirrored without rotation. The results

show that scenario 7, with only random mirroring, had the least negative impact on model performance, yielding a minor change of -0.39 percentage points in the weighted average mAP, and even showing a slight improvement for certain datasets. In contrast, scenarios 5 and 6, resulted in slightly larger decreases in performance (average mAP changes of -2.08 and -

2.45 percentage points, respectively). These findings suggest that especially the rotation of the images negatively impacts the model's performance, as it might introduce distortions.

The results of scenario 12, where we merged three frames into one to integrate a time component, yielded interesting insights. For datasets where the model already demonstrated robust performance, the accuracy experienced a noticeable decline. On the other hand, on datasets where the initial model struggled, an improvement of 6.2 percentage points was observed. This suggests that incorporating temporal information might be particularly beneficial when distinguishing between subtle features, such as pieces of wood and waves, proves challenging in a single frame. Further investigation into the impact of this temporal integration is needed to understand the specific scenarios where this approach is advantageous. These findings underscore the potential of leveraging temporal information to improve river wood detection.

Another adjustment in the method was evaluated in scenario 13, where we doubled the image size after rescaling. The scenario demonstrated its greatest improvements in performance on the three datasets (7, 11, and 18) that have the lowest relative bounding box sizes of the six representative datasets (see image 5). This indicates that the reduction in image size was too extreme, and samples can be missed. For custom detection algorithms, it is advised to calculate the relative bounding box size of the samples on their specific location and optimize the image rescaling in terms of performance and computational efficiency.

For the model's mean Average Precision of 61% on the test dataset (at the River Inn) it is essential to highlight that this accuracy was achieved despite the size of the Inn being larger than most rivers in the training database and the flood event's challenging conditions and the imagery's relatively low-quality nature. Images with dimensions of $1280 \times 720$ pixels were captured using a mobile phone in timelapse mode. Furthermore, it was found that the model is better at detecting wood samples with larger bounding boxes. The ability of the model to identify larger wood elements is essential for its practical applicability. Large wood components often constitute a substantial proportion of total wood transport within rivers (Galia et al., 2018). Hence, combining our deep learning model's proficiency in detecting wood facilitates quantifying wood transport in river systems. The results suggest that the model can be used to estimate and monitor wood transport dynamics in rivers, providing valuable insights into the ecological and geomorphic processes associated with fluvial environments. In cases with a particular interest in detecting the smaller samples, the limit in detectable size can be counteracted by increasing the image resolution or placing the camera closer to the stream.

### 4.2 Effect of neural network version on detection

The field of machine learning-based object detection moves fast. New versions of the existing state-of-the-art model are released every year. Therefore, we compared the performance of the 4th version of the You-Only-Look-Once model to the 7th (Wang et al., 2022) version. When comparing the training results on the same data using the base scenario, the results are shown in table 3. Even though the model became more efficient and smaller in terms of on-drive size (43% smaller from V4 to V7) and the resolution to which the images were rescaled was larger ($640 \times 640$ for V7 and $416 \times 416$ for V4), the performance did not drastically increase in our case. The average mAP went down by four percentage points, whilst the weighted average went up by 2.5 percentage points, mainly because the model performed better on the largest dataset. However, even though

**Table 3.** Comparison with YoloV7. The comparisons are made in terms of mean Average Precision at an Intersection over Union (IoU) of 0.5.

| Dataset No | mAP @ 0.5 IoU | | |
| | YoloV4 | YoloV7 | Difference |
| --- | --- | --- | --- |
| 1 | 80.33 | 77.18 | -3.15 |
| 7 | 71.17 | 78.53 | 7.36 |
| 11 | 43.67 | 43.92 | 0.25 |
| 13 | 92.83 | 90.29 | -2.54 |
| 18 | 35.33 | 21.07 | -15.26 |
| 20 | 57.17 | 44.82 | -12.35 |
| Average | 63.42 | 59.30 | -4.12 |
| Weighted Average | 66.41 | 69.01 | 2.60 |

the newer model is demonstrated to perform better on conventional ML benchmarks (Wang et al., 2022) and in specific cases also has a higher mAP in our tests, when using a model without finetuning it to a particular study site, the 4th version of the YOLO model performs better. The differences in performance between the models are greater than those in many of the training scenarios. Therefore, the model choice is still essential in developing a wood detection algorithm.

### 4.3 Understanding model predictions: wood features, surrounding water, and object size

The effectiveness of CNN has been displayed in various fields, such as security, transportation, and medical sciences (Kaur and Singh, 2023). However, even though it is not, the model is often considered a black box, so its trustworthiness can suffer. A CNN takes statistical relationships in the pixel data and constructs characteristics to infer floating wood. In our case, these characteristics are supposed to be characteristics of floating wood, like bark, root wads, branch stumps, and surrounding water. However, like a wolf-or-dog-classifier that was a snow detector (Ribeiro et al., 2016), it might use different characteristics of the training data to determine whether an object is wood. For instance, if a model is trained on data from a permanently mounted camera that constantly records the same scene, it can remember the scene and indicate everything out of the ordinary, like humans walking through the frame, to be large wood. If this is the case, the model would not demonstrate high performance on datasets that do not contain those characteristics. Therefore, it is essential to understand the model predictions.

We used one of the pictures of instream wood from dataset 20 found online to analyse which pixels in a figure were weighed the heaviest by the model to determine whether an object was a piece of instream wood. The pictures show a log stuck in a rapid, which has clear features of bark in brown colour with reflections and a fracture. Figure 8 demonstrates the inference of the image when using the base scenario as described in section 2.3.2 as compared to scenario 9. It also indicates the pixels the neural network uses to detect wood in the image. Remarkably, not just the pixels representing wood were indicated as applicable to detect instream wood. In this case, the training data contains almost exclusively pieces of floating wood, and pieces on the bank that were not floating were not indicated. Therefore, the network seems to require the indication of water-

containing waves next to the piece of wood to detect instream wood. In the base scenario, most training data contained small pieces of wood with a small relative bounding box size. Therefore, in the left image, the confidence of the model in the detected piece being wood is low, as the training data lacked a sufficient number of similar high-definition images. In scenario 9 however, high-definition images of non-floating wood were added to the training database and therefore, the inference yields different results. This image resembles the added images; consequently, the piece was indicated as wood with a higher certainty. Interestingly, the model seems to use pixels representing the wood (bark) texture and the fractured part for its detection. This means that it detects bark and fracture features, and these findings would underscore the hypothesis in section 3.2 that there was a delicate balance between wood detection and small-object (less defined shapes) detection, primarily driven by the average size of samples in the training data.

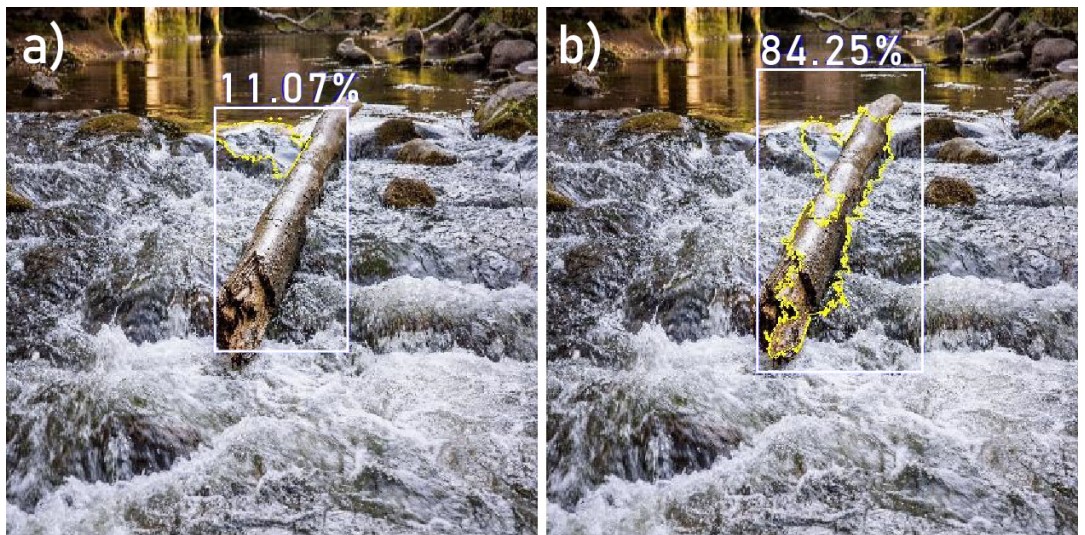

**Figure 8.** Wood detections according to the models trained with the a) Baseline scenario and b) Scenario 9: 'Added V1', where we added labelled closeup images of non-submerged wood. We indicate the heaviest weighting pixels in yellow according to the neural networks. Image source: dreamstime.com

## 4.4 Alternative Neural Networks for Wood Detection

While neural networks like R-CNN, SSD, and CenterNet offer potential for wood detection, each has limitations for our specific application. R-CNN's two-step process results in longer processing times, making it less suitable for real-time detection tasks. SSD can struggle with objects of varying sizes or those partially occluded, which is common in river environments. Similarly, CenterNet's key-point detection approach may not handle the complex and dynamic nature of floating wood as effectively. In contrast, YOLO's single-stage detection, with its speed and ability to handle diverse scenarios in real time, makes it a more practical and efficient choice for automating floating wood detection across varying riverine conditions. However, it is

important to note that we have not tested these models directly in our study, so their performance in this specific context remains speculative based on the general limitations reported in the literature.

## 4.5 Limitations of using low-cost cameras

Where low-cost cameras can aid research in offering economical means of capturing data, their use in detecting instream wood poses some limitations. Firstly, the lens and sensor quality of budget-friendly cameras often falls short compared to higher-end models, leading to less detailed images (Taylor et al., 2023). This lack of detail can make it even harder to distinguish small pieces of wood from noise within the frame (Casado-García et al., 2022). Additionally, the lighting conditions are generally handled less effectively, and glare from the water surface can obscure the visibility of wood. Lastly, in case of a lack of International Protection Rating certification, the lower durability of budget cameras in outdoor environments can lead to malfunctions and, eventually, gaps in the data. However, the benefits of data being widely available do make low-cost cameras a valuable and accessible source of data.

## 5 Conclusions

We trained a Convolutional Neural Network to detect instream wood with a weighted average performance of 67% mean Average Precision (mAP). On the best occasion, the model had a mAP of 93% on one specific dataset. The performance was sensitive to the quality of the images in the training data, as concluded by a wide range of results. On an unseen test dataset, its 61% mAP performance was in line with the results from the sensitivity tests. Efforts to improve the model's performance were, in some cases, successful. Depending on the data that was used for training, the model's performance increased by up to 23% (mAP). Changing the sampling strategy by adding or removing training data yielded considerable differences in average performance. Additionally, although enhancing the image input resolution increased the processing time and made the method more costly, in some instances, it did result in an almost 20 percentage points increase in mAP. On the other hand, data augmentation and different sampling methods did not seem to greatly influence the model's performance.

Even though it was attempted to create a training database with various examples, the training results still indicated the model overfitting the training data. Still, this study demonstrates that the model can generalise the concept of wood, mainly when training data consists of high-definition photos of labelled wood samples. Additionally, more fitting to the general applicability of the method, we show it can also generalize the concept of wood in rivers when the samples have different (smaller) dimensions. Large examples (of around 500x500 pixels) of wood were in the database, notably different from smaller samples (around 10x10 pixels). When training a custom model, it is advised to analyse the data that needs to be analysed and pick the datasets from our database accordingly. For this, it is crucial to use the training datasets that resemble it. A labelled training database of over 15,000 images was created in the research process. The training data is hosted publicly and can be used for future object detection refinements. Also, as the data is separated based on location and date, a customised model can be trained using the data that most closely represents the data of the person interested. For a new wood detection study, custom-labelled data can be added to the training database to increase the performance even more. This was underlined as tests

demonstrated that adding only 176 labelled images of the same monitoring station but on a different day could increase the model's performance by 19 percentage points.

Despite its potential, the proposed method can not yet be used in real time. In future efforts, smaller versions of the evaluated models, like the Tiny version of the YOLO model, could be developed to run on in-field or mobile devices. Merging three subsequent frames in certain instances improved results, suggesting that incorporating temporal imaging and the time component of a video could enhance the model's performance in detection tasks. Lastly, newly labelled datasets for custom models can be added to the larger database to aid in developing the performance of the model.

## 5.1 Recommendations for Future Development of (Custom) Wood-Detecting CNNs

1. **Tailor Training Data to Target Conditions**: For best results, use training datasets that resemble the intended deployment environment. For example, matching image quality, wood size, and contextual characteristics between training data and target conditions can improve performance.

2. **Prioritize High-Definition Image Samples**: Using high-resolution images can improve the model's generalization to the concept of wood, though this approach requires a balance with computational costs.

3. **Expand Training with Custom Labeled Data**: Incorporating additional labeled data, especially specific to the deployment site and context, can significantly improve model performance. For example, adding even a small set of labeled images from similar locations or conditions has been shown to enhance results.

4. **Consider Sampling Strategies**: Adjusting sampling strategies (such as including more or less training data) can impact average performance. Evaluate the trade-offs between model performance and data quantity when assembling training datasets.

5. **Investigate Temporal Data Integration**: Integrating information from consecutive video frames may improve detection by capturing movement patterns. This could be particularly relevant for video-based wood detection.

6. **Optimize for Real-Time Applications**: For real-time detection, consider experimenting with smaller model architectures, such as YOLO Tiny, to reduce processing requirements for in-field or mobile device applications.

*Code availability.* https://github.com/janbertoo/Instream_Wood_Detection

*Data availability.* The data to which we have the rights is available at: 10.5281/zenodo.10822254 .

*Author contributions.* Study conception and design: JA, VRV

Data collection: JA, MV

Methodology design: JA, TB, VRV

Analysis and interpretation of results: JA, TB, VRV

Manuscript preparation: JA, VRV, TB.

All authors reviewed and approved the final version.

*Competing interests.* The authors declare that they have no conflict of interest.

*Acknowledgements.* This work has been supported by the Swiss National Science Foundation project PCEFP2186963 and the University of Lausanne. The data from the Allier River has been made possible by Véodis-3D consultancy because of their financial and technical

assistance during the camera installation. We thank the comments from Prof. Iroumé, Dr. Diego Panici and Dr. Chris Tomsett, who helped us to improve the paper significantly.

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

## Appendix A: Training data analysis

### A1 Principal Component Analysis

The PCA analysis revealed a silhouette score of 0.034 with 6 clusters, using all 6400 dimensions of the cropped-out bounding boxes re-scaled to an 80x80 resolution. This low silhouette score from the PCA analysis suggests that our data is very diverse. In theory, this is advantageous as it suggests the potential for training a model to detect wood under varying conditions. However, distinguishing between wood detection and the detectionof less defined shapes depends heavily on the quality of the data used. The performance of a detection algorithm when detecting small samples can be compromised by including high-definition wood images, while the performance of a wood detection model can be impaired by incorporating datasets with small samples. Therefore, it is crucial to define the specific application of the model and develop a tailored approach accordingly.

Figure A1 shows the results of t K-means analysis. The fourth graph in the figure partitions the data into 8 clusters. One specific cluster exhibits a high silhouette score, suggesting a high degree of similarity among the images within this cluster. Despite efforts to eliminate duplicates, further examination of the data revealed that these images represent the same log positioned identically across successive frames. For future experiments, it would be advisable to remove the redundant instances in this cluster from the training dataset to enhance the model's performance.

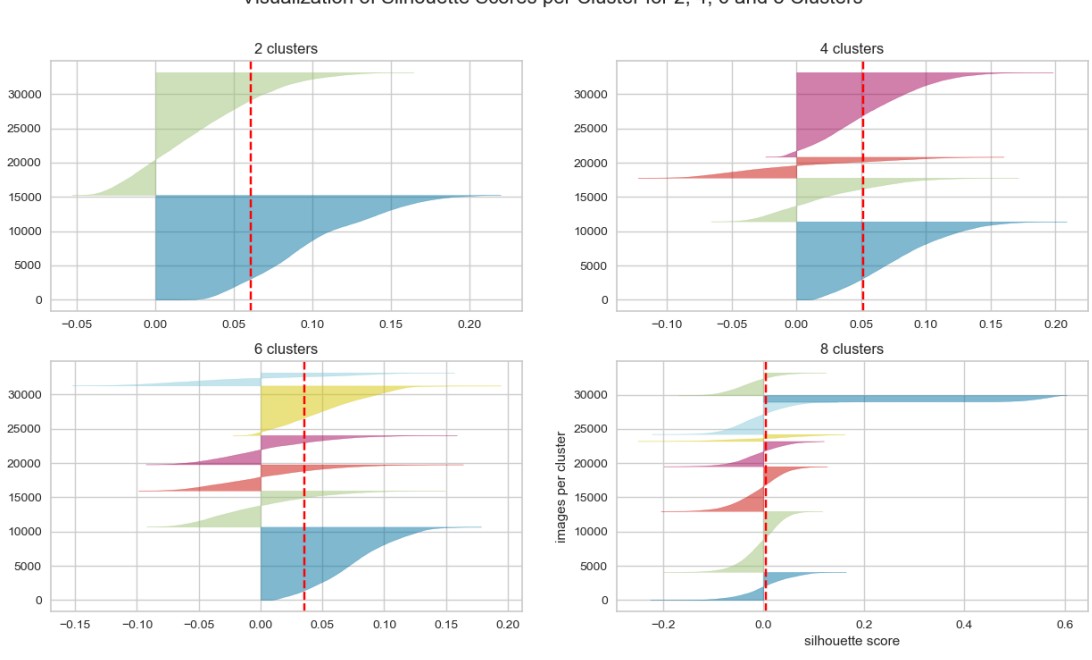

**Figure A1.** Visualization of K-means analysis with 2 clusters and a silhouette score of 0.061 (top left), 4 clusters and a silhouette score of 0.052 (top right), 6 clusters and a silhouette score of 0.035 (bottom left) and 8 clusters and a silhouette score of 0.005 (bottom right).

 **A2   Data diversity**

Figure A2 presents a visualization of the data diversity using T-distributed Stochastic Neighbor Embedding (t-SNE), a dimensionality reduction technique primarily employed for visualization purposes. The 20 known datasets are represented as distinct clusters, with each example marked by its cluster's colour. While the absolute distances between examples in the plot are not meaningful, the method does cluster similar neighbours closer together. The visualization demonstrates that, in general, the samples are well distributed. The overlap between clusters accounts for the low silhouette score, indicating high variability within the data. However, small, concentrated groups of images outside the central cluster can be identified as duplicates in the training data. To address this, the data-trimming step will aim to reduce the influence of these sub-clusters. This will prevent the final model from being disproportionately rewarded for correctly detecting a specific piece of wood, thus mitigating the risk of overfitting.

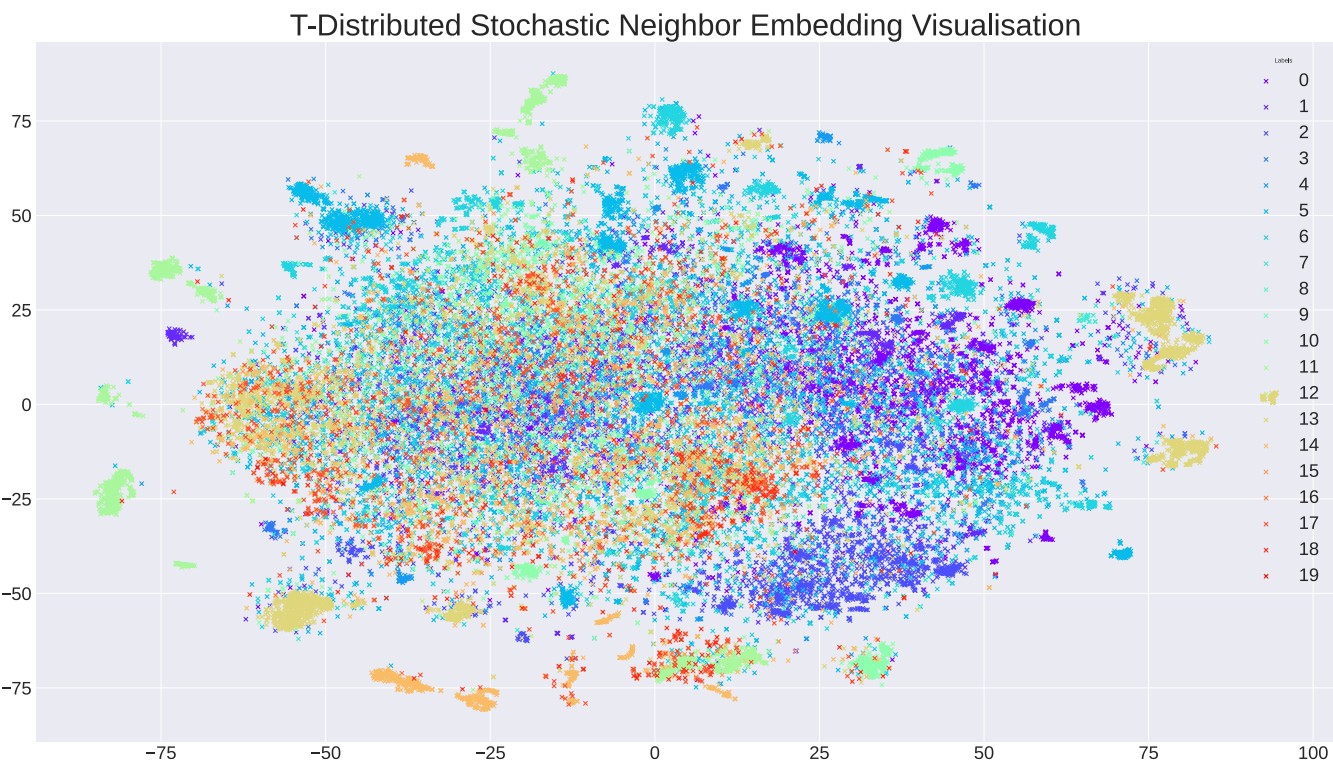

**Figure A2.** Clustering visualized using T-Distributed Stochastic Neighbor Embedding (Van Der Maaten and Hinton, 2008). Visualisation of all 33,000 samples. Different colours represent the 20 different datasets. Distinctive clusters in the figure have mainly the same colour and are, therefore, part of the same dataset.

 **Appendix B: Dataset acquisition example**

A variety of camera mounting techniques were employed to capture videos of floating wood, including using duct-taped mobile phones for stability in challenging outdoor environments. This allowed for flexible and accessible monitoring from bridges and stationary structures (see Figure B1). Figure B2 shows an example of the camera positioning at the Borgne d'Arolla. Cameras were mounted to observe the river from different angles.

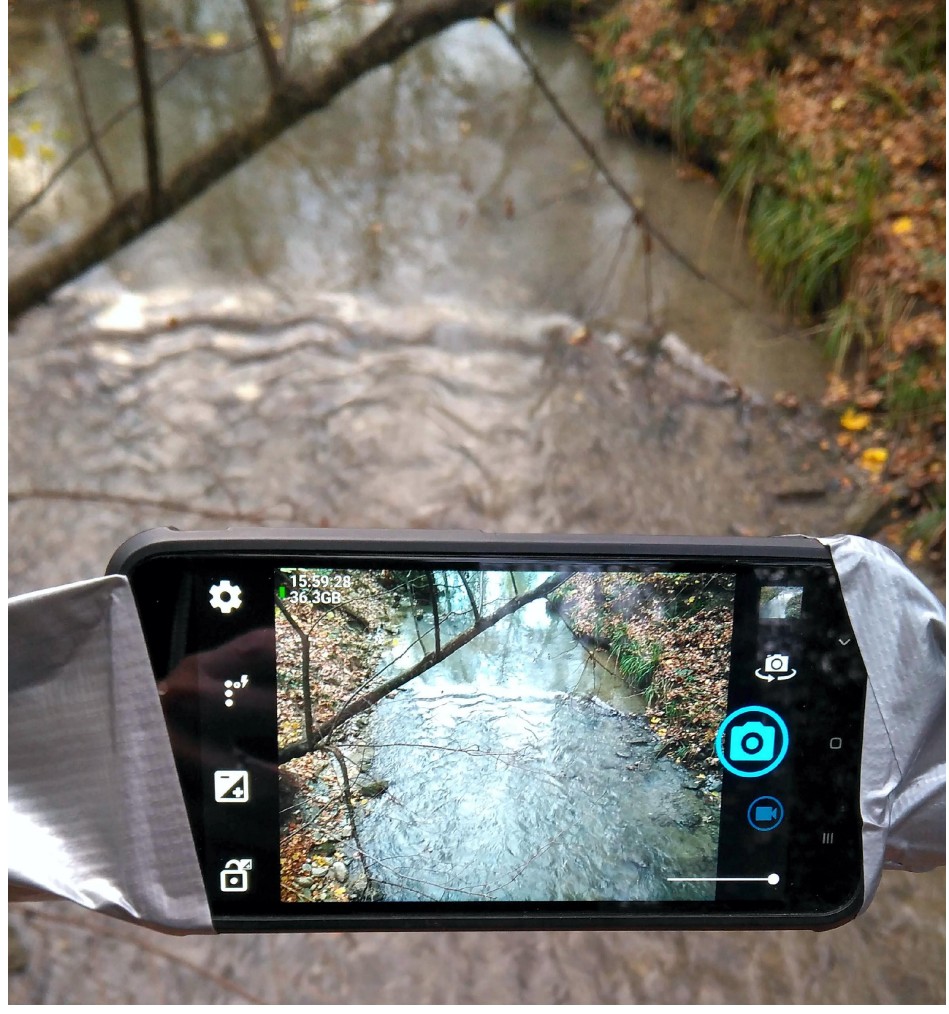

**Figure B1.** Example image data acquisition (dataset 5). A mobile phone camera was securely mounted using duct tape, providing a stable view of the river's surface from a bridge.

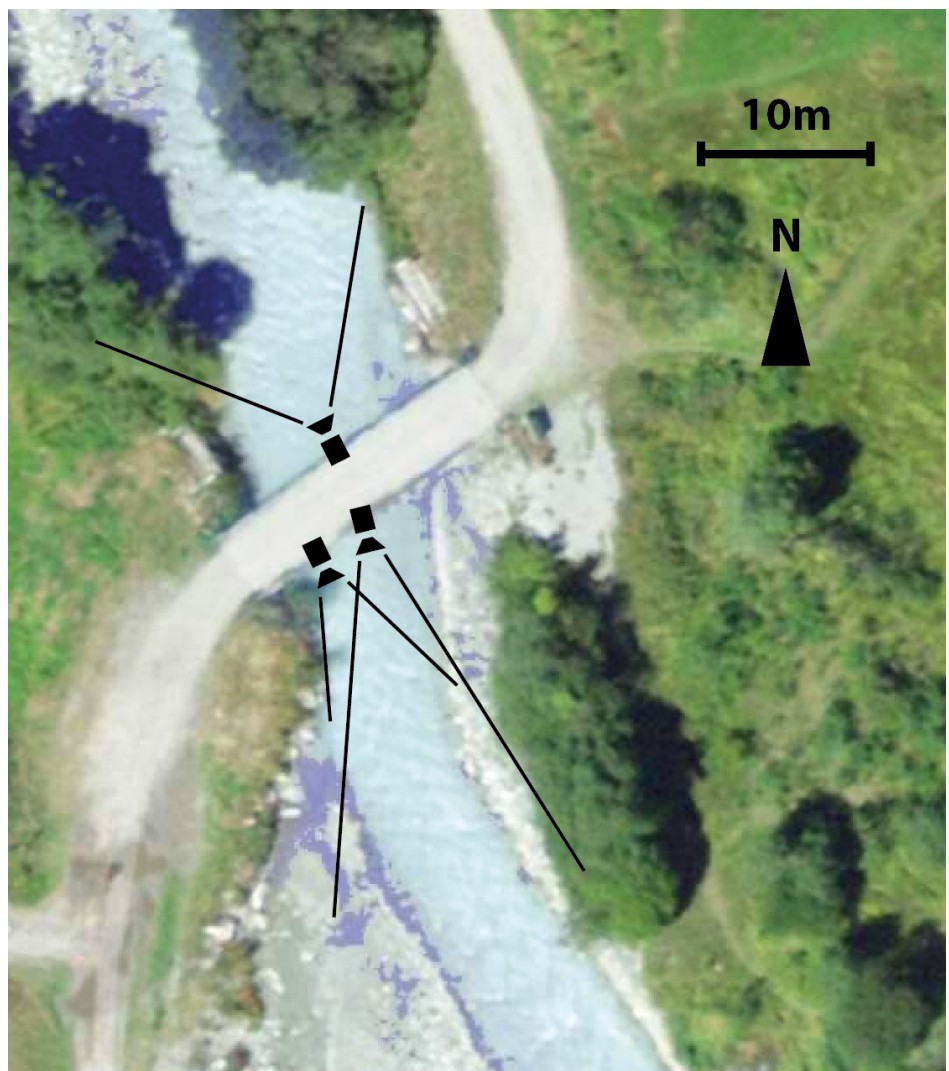

**Figure B2.** Example of the setup of cameras on a bridge at La Borgne d'Arolla.

755 **Appendix C: Dataset examples**

Figure C1 and C2 show examples of bounding boxes in datasets 1 and 12. The samples in dataset 12 are small and are therefore cropped and increased in size by 500%.

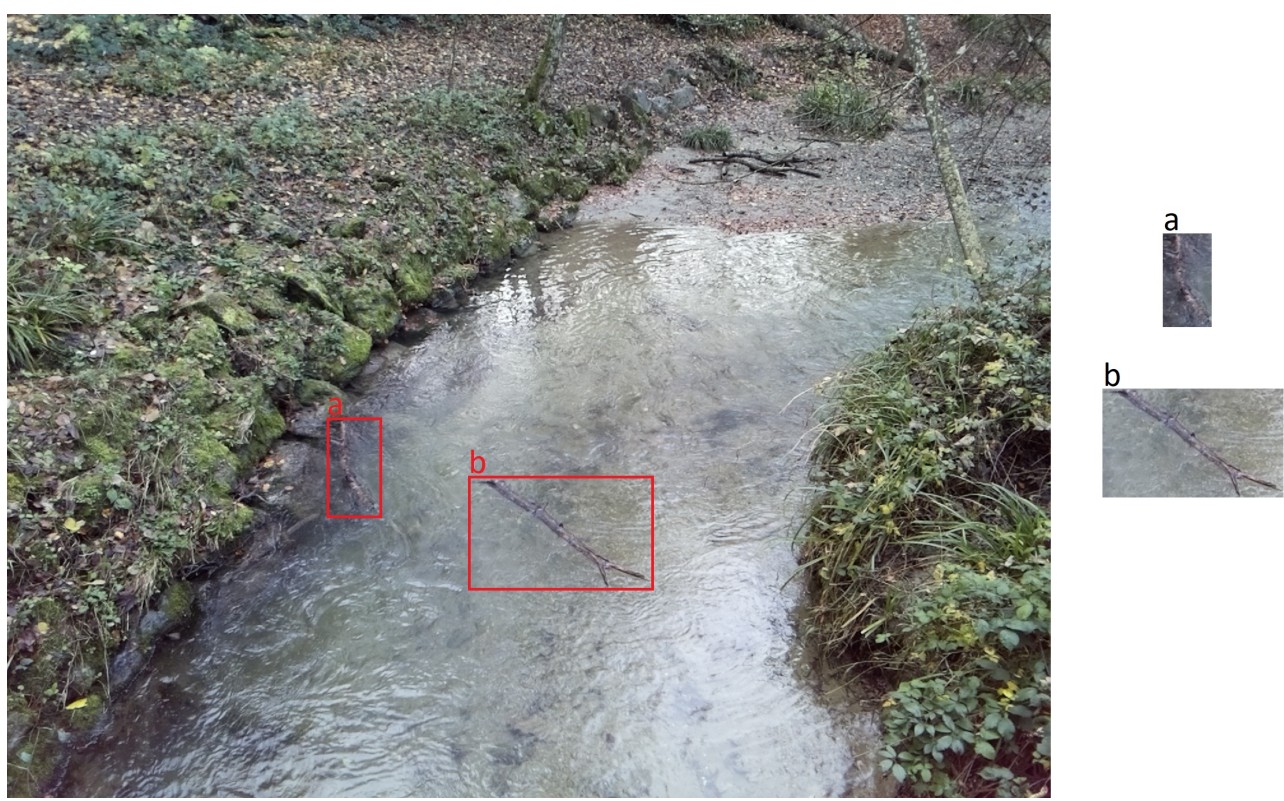

**Figure C1.** Example image dataset 1. The bounding boxes are cropped without resizing.

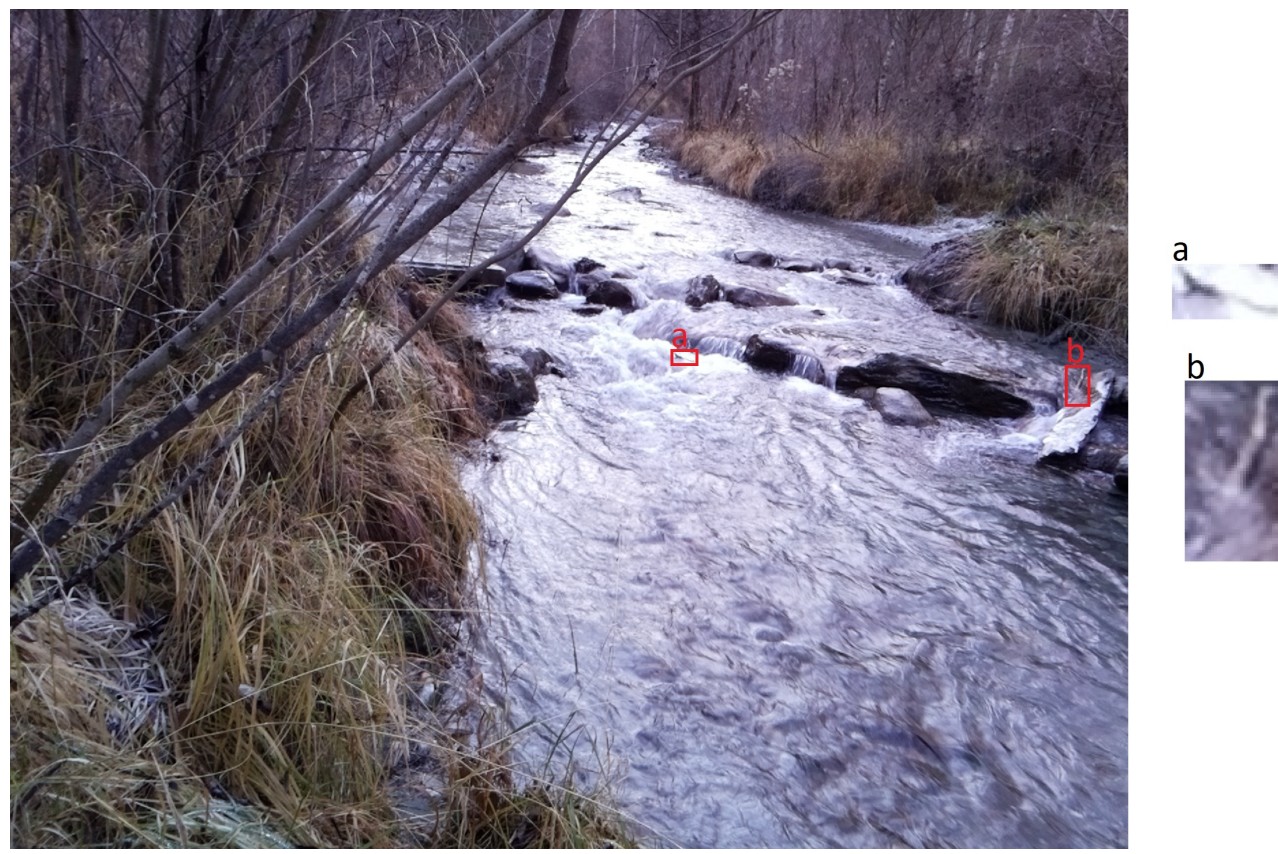

**Figure C2.** Example image dataset 12. The bounding boxes are cropped and uniformly resized by 500%.