# Peer review of "Automatic detection of floating instream large wood in videos using deep learning"

_EGUsphere, 2024_

## Referee Comment (RC1)

[referee-annotated manuscript omitted]

---

## Referee Comment (RC2)

**Overview**

This paper investigates a novel concept for monitoring wood in rivers, developing on existing algorithm development of CNNs for image recognition. There are numerous applications and possible impacts for this work, both for research and monitoring. The authors make a good case for the necessity of the research and offer a good grounding in some of the key concepts for a reader who is knew to machine learning methods, as well as those more familiar with them.

The writing needs to be improved throughout, as there are numerous occasions of informal writing which feel out of place, such as 'made a recent come-back'. Likewise, there are spelling mistakes and issues on consistency between American and English, which I am aware can be a challenge when writing in a non-native language. This can be helped by making sure both the spell-check and dictionary of all text in the document are set to one or the other. Moreover, some of the text struggles to convey the complexity of the methods in places, with repetition followed by missing detail.

The scenario design is clear, however how these scenarios fit in with some of the other analyses being undertaken is less apparent. There seems to be several sections which are additional scenarios/tests throughout the paper which are not clearly explained in the methods. Furthermore, there are numerous scenarios which are outlined in the methods which are not commented on in the results or discussion. These should either be discussed, or possibly removed (placed in supplementary) for the revision. Some of the additional analyses could then be incorporated as scenarios to make it easier to understand for the reader. Moreover, as the methods are quite complex, an improved schematic overview of the workflow would benefit the manuscript.

The results and discussion are currently presented as one. It would be best to separate them in this instance, with a discussion focussing on the reasons why some scenarios performed better, the limitations of the design, and the impact this may have for wood monitoring. This seems to be the largest element missing from this paper. Overall, it has great potential for helping to improve wood monitoring in rivers, but this is only briefly covered in the discussion, despite the overwhelming literature relating to the importance of wood in rivers, the hazards they present, and the different methods currently being used to monitor them. The results themselves are also not covered in full which is surprising as the order of magnitude of results is similar to those results that are covered.

The figures and tables throughout are of good standard, and with some adjustment would be suitable for publication. The only major differences would be the inclusion/adjustment of the methodology schematic as the visual elements would help the reader alongside the text, as well as the inclusion of a location map for the dataset origins in Figure 2.

Overall, the paper shows good promise and with some adjustments to the content, bolstering some of the justification, improving the writing standard, and focusing on the relevance of the work, would provide a useful addition to the field.

**General Comments**

| Section | Comment |
| --- | --- |
| Introduction | The opening to the manuscript is very clear laying out both the justification for this research as well as introducing concepts around the importance of in-channel vegetation and the anthropogenic forces governing this area.

Some aspects could do with a greater reference to existing literature, as well as making sure to give examples when the authors make certain statements regarding current methods and results.

Take care also not to generalise fields of research too much, to state the wood transport data is scarce seems a stretch, as there are examples of research into this. Perhaps rephrasing these types of statements to be softer would be beneficial, even with the caveat of this being a growing area of interest.

Some of the latter paragraphs don't tend to flow well between one-another, and potentially some better link statements and reordering would help this feel less 'chunked'. Some suggestions provided herein.

The final paragraph is almost to detailed for the introduction, but not detailed enough for the methods. I would suggest removing part of this (line 67 onwards) into the methods section to help justify your choice of CNNs. The new last sentence would act as an aim of the paper, and as such, a few points to act as objectives could help round of the introduction nicely. This is currently not clearly stated and so would need to be included, even if not explicitly listed. |
| Methods | There seems to be a large omission of background into the different possible algorithms that could have been used (both CNNs and other) at the beginning of this section. With the addition and expansion on some of the text from the introduction, as well as more explanation into how they work and why they are suited for instream wood detection, this would be more suitable, especially as this paper feels like it is trying to reach both those already involved in machine learning methods, and those who want to detect wood better but do not have the experience. Likewise, the choice of YOLO as an algorithm is not fully justified in the text, and so could do with some more supportive reasoning here.

Again, there is some use of informal phrases, such as 'the goal is to have a lot of diverse data', whereas, 'CNNs require large volumes of diverse data for effective training', would be better suited.

Overall, there is a tendency to omit some key pieces of information from the methods as outlined in the specific comments below. These stop the reader from fully understanding the methods that have been undertaken, and without them would not be suitable for full replicability.

The overall layout of the methods section is well though out with clear subsections to help guide the reader through what is clearly an extensive |

| | |
|---|---|
| | piece of analysis, but the detail makes it hard to truly understand what's going on in places.

Although the splitting of the data into train, validation, and test, is somewhat stated at the beginning of the methods, I don't think it is overly clear that this is done on a dataset basis for each of the following scenarios. I think some clearer wording and phrasing here would help the reader to know exactly what is being used as training, validation, and testing data. Again, I am not sure if a better overall methodology schematic would be suitable, building on figure 1 primarily.

The scenarios used for assessing model development are in themselves valid, but seem to place arbitrary values and not investigate the scale over which these operate. I.e. a minimum of 500, but does not overly look into the effects of changing this arbitrary value, which could have better answered a few scenarios in more detail. This is partly answered in the sensitivity analysis, but using two values does not really give a sense of how the ability of the network to learn with different numbers of training images develops. I am also not sure the sensitivity analysis is really that different from some of your scenarios, and as such maybe rolling them into one section is better here. When outlining your scenarios, you also sometimes state the number of images altered, and how this effected training dataset size compared to baseline, but not always, even for similar scenarios. Try to be consistent in the information provided.

Overall, there is a lot of key information in the methods, and it is relatively well laid out. However, the discrepancies between information given, and a lack of lineage in terms of going from raw data to a final test of an optimised model could be improved. Certain sections seem out of place, with some underdiscussed. The methods themselves seem sound and novel given the prior work in this area and could make a valid contribution to the research in this field which is great to see. |
| Results/
Discussion | I think there is a need to separate out the results and discussion. Although in some cases a more fluid approach is justified, in this scenario I think it becomes a little unclear as to what you have found and what you are discussing. This links to your section headers. The methods has sensitivity analysis and the scenarios as separate, but your results combines these. I think it would be wise to align the headings so each one refers to each section of the methods, this will help the reader.

From doing this, it should help to solve a follow up change, whereby your headings for each part of the results/discussion summarise your findings. I really like this idea so the reader has a clear description of what the next section is discussing, these could work nicely in a separate discussion section along with some other suggestions further in.

I am unsure whether the PCA is adding much to the argument here, a lack of access to supplementary to check this means further comment cannot be made. However, figure 5 shows quite similar datasets on the whole with a few outliers, contrary to the narrative. |

| | Your results show the outputs of several augmentation scenarios and sampling scenarios (1-8), but these are barely covered in any results or discussion, with most of the writing focussing on scenarios 9-13, which begs the question of why these scenarios were done or even included to begin with? The values in table 2 suggest the changes in model outputs are not drastically different than those that are discussed either.

The end of the results/discussion section feels a bit rushed at points and does not really go into some of the detail on interesting questions surrounding model selection and the test dataset. I am also unsure of the comparison to an extra downloaded image of wood in rivers, as it seems quite detached from the traditional imagery which is primarily captured in these scenarios shown in Figure 7. |
|---|---|
| Conclusion | Overall, the conclusion is well written and summarises the results nicely. I think the assumption that the model does not understand 'what wood is' until you add some extra photos is overly critical, as the model is understanding 'what wood is' when transported and observed from a bridge or other viewing point. Finally, the next steps could be clearly grouped into some neat bullet points to show where the research should be heading. |

**Specific Comments**

| Section | Line | Comment |
|---|---|---|
| Introduction | Lines 19-20 | There are quite a few more important sources of large wood, such as windthrow and natural mortality, or influence from fauna. |
| | Line 25 | I think localised rather larger is more appropriate here when talking about inundation. |
| | Line 26 | Could likely do with some more up to date references here to show advances in this area. E.g. https://www.mdpi.com/2076-3417/13/18/10454#B65-applsci-13-10454 and https://www.mdpi.com/2077-1312/10/7/911 |
| | Line 36 | Although this is introductory, I think some reference to those existing algorithms, and why they are location specific would be beneficial, especially as one of the goals of the paper appears to be to reduce the site-specific nature of current wood detection algorithms. |
| | Line 40 | Comma needed after high-resolution aerial surveys. This occurs in other places where you are listing. |
| | Line 41 | Give some examples of how RFID is being used, such as https://onlinelibrary.wiley.com/doi/full/10.1002/esp.3463?casa_token=G-p1V7DmbDEAAAAA%3AcGP5b3hqKzlPyE8YEHHSPK78ppWrXGMST7iPG-JZUsnuplmrtM2Vs6gkX-LlQYRsjPiCq-bqfgCXEA, https://onlinelibrary.wiley.com/doi/abs/10.1002/esp.1888?casa_token=w1NFWrbZJ7gAAAAA:0YvqxuFyU7vHaDZ2FQ3hHxIDP474jAXCKdoHvR_oKZbKkLphb0btepE7Yw0yjn9ZpJW3KPwQb6tyYQ, and the authors prior work using RFID to improve CNNs. |
| | Line 54 | Introducing a sentence along the lines of 'they are also limited by their spatial locations, and rely on specific setups being installed prior to an event' This would lead nicely into the use of citizen science. |
| | Line 61 | Similar to the above, another link sentence here would help the flow, think along the lines of 'Advances in machine learning methods may help to overcome this and allow for widespread wood detection'. |
| | Line 62 | I am not sure this first sentence is needed, feels informal and unnecessary. |
| | Line 67 | Starting 'The CNN has multiple…', move this to next paragraph in methods and needs expanding (see comments relating to this below). Then add in some objectives as to how you plan to run, test, and evaluate your algorithm development. |

| Methods | Line 78 | The first sentence explaining your choice of YOLO algorithm should be proceeded by a small review (one paragraph) on how CNNs work, and why they may be more suitable for detection than other algorithms, building on the information from the introduction. |
|---|---|---|
| | | Following this, I feel that just saying YOLO was chosen for speed and accuracy is limited, especially as this is tested on generic imagery in their paper not large wood. Is there any reason to suggest it would be better for large wood? If not, have other studies trying to detect wood compared between algorithms? Think this is quite a crucial area to justify. |
| | Line 80 | With above changes, a new paragraph could likely be started with 'Training a CNN…' |
| | Line 86 | Combine these two sentences for better flow. |
| | Line 89 -95 | I think some more details about the quality of the cameras here would be useful, such as resolution. Do you have a record of how much wood was added to each stream? How long have the monitoring programmes been underway in France and is any of that manual input to the channel? Where are the online videos from and what helps to make the images and wood a more diverse setting? These are all questions that need addressing. Does each image represent a single piece of wood, or are there more pieces of wood in each image, relating to the 15,228 number here. I think you should refer to table 1 here, and also adjust figure 2 as outlined below in figures and tables section. |
| | Line 98 | Figure 2 also shows bounding boxes, maybe reference this instead. |
| | Lines 99 – 105 | This section could do with some additional clarity, especially as this is an additional CCN alogirthm being deployed I assume? Were you checking that the automated bounding boxes for these were detecting wood, as this is not clear which dataset you are referring to by saying the labels were checked manually. There is also no explanation of why this worked better for 11 of the 15 datasets, or what your tolerance for acceptable mAP was, especially as not good enough was below 20 percent. Moreover, when images were checked manually, were incorrectly labelled frames eliminated or adjusted, or left as incorrect? |
| | Line 112 | Why 80*80, is this purely incidental that no wood was larger than this, I also assume this is in pixel size not other units? |
| | Line 116 | There is no statement of why this PCA was undertaken, and it only becomes clearer when reading the results. You need to add some context as to why this is undertaken. Furthermore, if the results of the t-SNE test are stochastic, could you not |

| | | run the test numerous times to assess the diversity, akin to monte-carlo scenarios? |
|---|---|---|
| | Line 125 | Swap must for 'is typically', if smaller datasets don't allow it, there is not always a split in this fashion or a separate test dataset. |
| | Line 125 – 142 | This section is trying to explain a somewhat complex training and validation procedure, whereby computational trade-offs mean omitting some of your data as validation. However, it feels as though how these 6 examples were selected is not overly clear, besides not being at the same place and time. It may have made sense to use dataset 14 also, purely as that would give you validation samples at a range of sizes.

The section took a while to become clear as to what the process was, and that datasets weren't being dropped from training, just the number of validation sets dropped. Perhaps trying to simplify the wording in places and go through the order. For example, 6 validation cycles were run, for each one a single dataset was dropped for validating and the model trained on the remaining 19. These 6 were chosen to represent a range of conditions, and reduced computational overhead by not undertaking 20 validation cycles. |
| | Line 139 | Where has this extra dataset come from, and why was it not introduced with the other datasets? Who has been studying this, a research group(s) or monitoring agencies?

This is a useful case example that in essence the paper could have been framed around. I.e. instead of can we implement a cool algorithm, can we reduce human labour of monitoring wood? |
| | Lines 145 – 152 | This seems to be an odd way to do your sensitivity test, as although you are trying to identify the effect of number of inputs on output quality, if these inputs are multiplied for smaller datasets, then they are not adding any extra information, only overtraining the model? Would it not have been better to undertake this at a smaller number of images to assess performance, on possibly a limited number of datasets. I.e. for 10 of your datasets over 300, or 8 over the 1000, sequentially go from including 100, 200, 300, etc, and then quantify at what point there is no improvement in the model? This in itself could be one of the scenarios. |
| | Line 154 | Although it is clear there are no river based large wood studies to learn from, it seems that casting the net a little wider shows these studies have been used in similar ways on living trees and perhaps other wood related scenarios. E.g. https://ieeexplore.ieee.org/abstract/document/9643113?casa _token=Vm749u_aLtQAAAAA:lQ8hGqEscq00Tf4M5Co8uVAJ1q siJtDGUoMrQDFj-oSM14tTKiVBbKzIUl1G00TwZ5AGgRy_qw |

| | Line 158 | Although in principle I can understand how all these parameters effect wood detection, but has any worked actually been conducted on this? If so, reference it. |
|---|---|---|
| | Lines 158 – 160 | It says 14 were trained and compared to a baseline, but there are 13 outlined. Either say 13 models trained, or 14 including the baseline for which the other 13 are compared to. |
| | Lines 162-163 | Why were the values of 4% and 30% chosen, is there a rationale for this? The logic behind this makes sense, but just need to clarify reasoning for thresholds, even if they were just decided as no previous study to base upon. |
| | Line 170 | Can the dataset size be given for total number of images, i.e. how similar is 'approximately the same'. This also feeds into informal language comments. |
| | Line 171 | Why such a high number, when lowering it slightly could reduce the need for oversampling from some datasets? This appears similar to the sensitivity analysis you performed. |
| | Lines 176 – 179 | Can you specify the number that were rotated vs mirrored, as the and/or makes it unclear if this was randomly done and randomly distributed. Were any both mirrored and rotated?

I agree, that only partial rotation is necessary, this seems like a sensible decision to have made. |
| | Line 180 | Again, how many were altered, and what proportion were mirrored or rotated. I think this needs more detail so the user knows what was done. How much extra data did this result in? |
| | Line 188 | I feel that the inclusion of the phrase 'non-living wood' implies you are adding living wood, as opposed to wood that is not floating. Could be removed. |
| | Line 189 | Change example to wood sample. |
| | Lines 192 – 196 | This is an interesting scenario, primarily as these are open source datasets, with lower quality, but greater geographical diversion. In essence, I am not sure this is just testing data quality. Again, with the addition of other datasets, I think they should be mentioned in the original introduction of data, and their locations (even if approximate) included on a figure map. They can be highlighted/commented that they are only used for testing or specific scenarios, but curious as to why they were not included from the outset? |
| | Line 199 | Which datasets were removed? |
| | Lines 201 – 206 | This is really well explained and justified here, so should therefore be a model for your other scenarios where the justification is weaker. |

| | Lines 207 – 211 | This is an interesting scenario to assess, as many secondary data sources may be of lower quality. However, are the double-precision images used either a) the down sampled images at 416*416 resampled again to a higher resolution, or b) the original images resampled to 832*832? The text make it seems like you double the resolution of the down sampled image (scenario a), as opposed to changing the original resampling (scenario b). Make this clear either way. |
|---|---|---|
| | Lines 213 – 226 | This is a really well written section on the statistics being used, what they mean, and how a reader should interpret them. |
| | Lines 232 – 239 | How is this sensitivity test different to the one introduced earlier, and why has this one got more dataset sizes to test the sensitivity? This is also not referred to in the results as far as I can tell, so what is the purpose of this section?

The variance method also adds some confusion, is each of these models run several times, and the best results taken? If so, why the best results, does that not overestimate model performance? This could be explained better.

You then talk about comparing between models, which again is fine but is very brief as to why, needs more explanation. You then mention a final model, is this not just your optimal model from all your testing? |
| | Line 238 | Which dataset is this, is it the same as the one introduced previously on the river Inn? Again, this needs to be stated. |
| | Lines 240 – 244 | This seems like a really sensible addition and is good to see some unpicking of what is happening behind the scenes. Maybe a brief idea of how this works, and what you hope to find and why you picked certain images (one river, across rivers, different angles?) would be sensible? In this case you might hope to hypothesise why some images/datasets are less well classified? This would be nice to see expanded on in the discussion. |
| Results/ Discussion | Line 248 | I am not sure 'blob' is appropriate here, and if they are so small how can you be certain these are pieces of wood? You mention wood remaining stationary, does that mean moving wood was not included in the study? |
| | Lines 254 – 264 | Unfortunately, no supplementary could be found on the online interface for comparison. However, I do wonder if whether double panelling a figure to include one of these plots for clustering with figure 5 could help to show the variation.

I would also argue that the relative sizes of the bounding boxes compared to images were not that different, with many similar distributions and a few outliers, primarily from external datasets which is to be expected. |

| | | |
|---|---|---|
| | | You also state that for 12, 18, and 19, the drop in relative size could be due to low camera resolution or distance from stream, but 12 is one of the model setup cameras so surely you know this, and could tell for the others by looking at the original images? |
| | Line 265 | Assume this is meant to be Database Configuration… |
| | Lines 266-270 | As this is both a results and discussion section currently, there is a lack of discussion here about why this may be, and that by oversampling images you may not see an improvement in model performance purely due to the model become more tuned to those specific examples. |
| | Line 270 | This is very important, if you do not now oversample, in your scenarios where you mentioned oversampling smaller datasets, did you now not do this? This seems like quite a big change. If so, I think the sensitivity results need to come within the methods inclusive so that you do not explain changes in your methods during the results. |
| | Line 273 | What were these results, and are they really comparable considering the differences in the object types? |
| | Line 274 – 276 | This section is not overly clear, I think it needs better wording to explain what is being done here, especially regarding the multiple training rounds. This feeds back into above comments at the end of the methods. |
| | Lines 281 – 284 | I can see what is trying to be said here, about training for specific or general wood detection, but feel it could have been said better. This is also the first mention of how cameras were mounted, perhaps this should be mentioned in the data section also. |
| | Lines 285 – 293 | There is a focus here on the high-definition wood images in this analysis, and yet there are only 9 images in the dataset. As such, are larger changes in mAP not more likely due simply to the lower number of objects to compare against? This is somewhat shown by the weighted average, and so overstating the importance of a vast performance decrease or increase here may be unjustified. The narrative however, that good wood images lead to better training than poor wood images, is justified by the average and weighted average outputs. |
| | Lines 294 – 295 | Has a significance test been undertaken here?

Are these broadly speaking not the only two factors, apart from manual labelling to begin with for training.

What are the worst performing models? |
| | Lines 296 – 297 | This sounds like you have added in an extra scenario, rather than describing one of your scenarios. |

| | | Change 'where the datasets with lower performance than 30% mAP were excluded' to 'where the datasets with a mAP of lower than 30% were excluded'. |
|---|---|---|
| | Lines 300 – 301 | Which scenario is this, can't find a reference to 19% in the table that is positive? If this is just assuming the inverse, then the addition of these images back wouldn't be the same 19% as the base conditions would be a different value. |
| | Lines 301 – 306 | This is a really important and useful point, and should be one of the key take home messages that adding to existing databases with some data from a site improves the algorithms performance. Check some wording here though, especially when speculating performance benefits. |
| | Lines 307 – 313 | This is an interesting section about whether the time component is critical. However, I fell it is overplayed in its significance. Of the two worst performing datasets (11 and 18) only one shows an increase of 6%, the other a decrease. Therefore, to say improvements of nearly 10% are made is an exaggeration. Arguably, this is somewhat upstaged by the large decrease in one of the better performing datasets (3). |
| | Lines 313 – 316 | Make this a separate paragraph as it feels separate from the temporal component.

Compared to the emphasis placed on scenario 12, scenario 13 appears to show much greater performance gains, and the importance of image resolution in tracking wood. As this has implications for how wood should be monitored, both from a hardware and software perspective, it likely needs more attention and discussion around the trade-offs between image resolution, computational efficiency, and expected wood size.

Line 315 references image 5, is this from figure 2 as these seem to be larger wood size, if not, please be clearer as to what this refers to. |
| | Lines 317 – Onwards | This almost feels like a different section or subsection, as it is a change from training and validating to assessing the model used. It seems as though this section itself however is limited in just comparing two models, moreover, these results have differences greater than many of the scenarios provided above, which indicates that model choice may be more important than datasets, something that is not discussed in great detail. As a result, the take home would switch from the importance of data, to the importance of model selection in getting the best outputs… |
| | Line 329 | Perhaps, if a new subsection is introduced for the above, this should be moved prior to it. |

| | | |
|---|---|---|
| | Line 334 | Reference figure 7 here, as it is not referenced anywhere in the text |
| | Lines 335 – 337 | You identify that the model is better at identifying large wood, and then state how large wood components compromise the greatest proportion of transport, but this needs to be referenced to support this. Furthermore, small wood components also play a role in increasing the total volume of log jams etc and so important to monitor. Commenting on how this is missed in the dataset is probably needed.

If possible, it would be great to look at those that are missed and estimate the size of these to identify a limit of detection. However, that may be beyond the scope of this investigation and potential for future research. |
| | Line 338 | Have these images been georectified in the processing? If so this needs to be explained for reproducibility. Moreover, if they have then they could be used to identify the limits of detection for wood as per above? |
| | Line 342 | Give examples here please, and comment on how they may differ or align to wood detection (e.g. shape and background). |
| | Line 342 – 346 | I think this needs to be reworded, at times this sounds speculative and also non-scientific. The theory of not being able to detect outside of the training sample is sound, just the transmission of this information is not clear enough. |
| | Line 347 | Where was this from and why not use one of your current data? Again, this points to questions going back to your initial data introduction, and consistently adding new bits of information. |
| | Lines 350 – 360 | Does this not come back to simple survey and image design. If most of your images are from roads and bridges overlooking rivers, and you provide an image much closer to the channel, it will struggle, until as you say you include images of large bits of wood close up. Therefore, to use the word remarkably again seems a little overstated. |
| | Lines 357-358 | Can you expand on how you know it is using the wood texture, is this hypothesised from the location of the pixels used, or can this be proven? |
| Conclusion | Lines 363 – 371 | This is a nice start to your conclusion, summarising your results well to give an overview of the paper. However, there is no comment on how increasing data sizes or changing their angles/mirroring had no effect. |
| | Lines 372 – 382 | I feel that to say your model struggles on the definition of wood, unless its given high-quality images of wood not in rivers, is overly harsh on your model. The purpose of this paper and method is to detect wood in rivers, likely from monitoring stations above the rivers surface (on bridges etc). So the |

| | | model works if it detects these well, and shouldn't necessarily be able to detect wood such as in Figure 8. Therefore, the model CAN generalise the concept of wood 'in rivers', which is the main purpose is it not?

I think the word blob should be removed throughout, perhaps in this instance they are best referred to as fragments or segments, i.e. not all the wood is on show? Make sure this distinction is first explained when replacing the initial occurrence of the word 'blob'.

This may be clarified by an earlier point, is this 19% increase simply the opposite of the 19% reduction when the Allier dataset (18) is removed? If so. This is not 19% (e.g. 20% decrease from 100 is 80, a 20% increase from 80 is not 100). If this is a separate analysis, make sure this is clear during the methods and results. It could even be viewed as an additional scenario (e.g. adding same site from different date). |
| | Lines 383 – 387 | This could likely be grouped into areas of future research. 1) real time monitoring 2) algorithm development and miniaturisation 3) temporal imagery for object detection. These could also form some structure for a separated discussion, allowing room to discuss the impacts of the research. |

**Figures**

| Figure | Comment |
|--------|---------|
| Figure 1 | This figure could benefit from labelling the boxes with the sections of the method that they refer to. This will allow readers to quickly understand which bit of the process they are referring to. Make sure the naming matches to, it will help the reader.

This could also be improved by creating this as an overall schematic of the methods, which would better describe the whole process as mentioned prior. |
| Figure 2 | It is great to see some visual examples of what these images look like, and how they differ, especially in regard to the additional imagery. However, I think it would be good to possibly remove one or two images, and add an inset location map showing where in the world these were taken from, rather than coordinates in the caption. This would give a better idea to the readers of where your data is coming from. You could colour or size location dots based on the number of images from a location as well. |
| Table 1 | Could this table also have a column or some stars which denote the datasets used in validation, these are mentioned later on but will help the reader when scanning back and forth. Consider making either camera lowercase, or the unknown and differing upper case. |
| Figure 3 | Why is this figure not further up in the manuscript? It is referenced first several pages earlier and causes confusion in the current section. Appreciate this may just be a current formatting error for the preprint. |
| Figure 4 | No changes required for this figure, it is clearly laid out, shows the size of datasets, and helps to explain what is happening in terms of the number of training vs validation datasets. |
| Figure 5 | Again, another clear figure which adds to the manuscript and is broadly easy to interpret. The inclusion of a double headed arrow along the x axis, pointing to larger wood and smaller wood may help with interpretation, so readers know if the value is indicating a lot of the image is the woods bounding box, or little. |
| Figure 6 | This figure is good, however it could do with stretching along the x axis, as this will help to show the variation in IOU training loss which show subtle differences. |
| Table 2 | The table layout is fine, but the text is a little hard to read in places. For those reading in non-colour or with colour-confusion, perhaps as well as colours a marker could be used to quickly attribute greater than 3% increases or decreases. |

| Figure 7 | A useful figure, make sure it is referenced in the text. Are these bounding boxes ones predicted by the model or drawn manually for users. It could be better to include boxes created by the model as well to show the types of wood it is missing (perhaps detected and missed wood as two separate colours?). |
|---|---|
| Figure 8 | Are the bounding boxes in this figure manually drawn? If so, they should probably better align with the extent of the wood. Likewise, as the percentage is referring to overlap in bounding box size, perhaps indicating the bounding box of the detected wood would help to illustrate these differences? Otherwise, this is a very helpful and useful figure. |

---

## Author Response (AR1)

*Dear Editor and reviewers, thank you very much for revising our manuscript. We appreciate the time taken by each reviewer to revise our manuscript and the suggestions that helped improve our work. In the following, we reply to each comment. We made all changes accordingly in our revised version. Lines here refer to the revised manuscript with tracked changes.*

*Janbert Aarnink on behalf of the co-authors*

**CC1 Comments by Prof. Andres Iroumé**

Is a very well written and interesting manuscript.

I have a few suggestions intended to complete/improve some aspects.

*Response: Thank you for your comments and help in increasing this manuscript's quality. The suggestions were well appreciated.*

They are:

Introduction:

- Page 1, L19-20. Natural mortality wind, snow loads, wildfires and beaver activities can also be recruitment sources.

*Response: thank you. The reviewer is correct; we added the abovementioned processes in Line 18.*

- Page 1, L20. "Wood plays a crucial role by trapping sediment, creating pools, and generating spatially varying flow patterns", not only as it distributes along the riverbanks but also when stored within the active or bankfull channel.

*Response: we edited the sentence to clarify this aspect in Line 19 in the revised ms.*

- Page 2, L34. The number of observations of instream wood is scarce? I do not fully agree. Perhaps the amount of observations of instream wood dynamics is scarce, so please clarify.

*Response: The other reviewer also raised this point, so we edited this paragraph to clarify what we meant. Lines 35-37.*

- Page 2, L43, about the best methods to quantify wood transport. Not only video-based methods, but also the installation of a GPS in each wood is a very good method, but extremely expensive.

*Response: yes, agreed. We added this and other approaches in the revised text. Lines 37-45.*

 Methods:

- Page 3, L86. Figure 1 does not give an overview of the data collection and processing. It gives an overview of the process to follow to collect and process data. Please also correct the title of Fig. 1 below the figure.

*Response: we corrected the text accordingly. New caption of Figure 1: Overview of the methodology used for data collection and processing. .*

- Page 4, L107 and 115. Figure or figure? Please decide.

*Response: we corrected the text accordingly across the manuscript.*

Discussion and conclusion:

- I do not find comments related to the limitations of the use of low-cost cameras, and how to avoid these limitations, may be by using high resolution cameras, installations, others. Please discuss and conclude.

*Response: yes, this is an important point. We added some discussion about the use of high-resolution cameras. Section 4.4.*

**RC1 Comments by Prof. Diego Panici**

The manuscript is about the automatic detection of instream large wood in video recording using deep learning tools. The results are really intriguing, but I believe that a substantial revision will be needed before considering this paper for publication. Here are some major comments:

First, there is limited to no comparison with other existing models. CNNs are widely used for image recognition (and, indeed, the authors acknowledged YOLO being the most widespread algorithm), yet, there is no comparative analysis with other studies or algorithms.

Second, the overall aim and output of this manuscript is really unclear. It is necessary to explicitate this further and emphasise what the study has revealed and what increase in scientific knowledge it has brought. As things stand, it is hard to discern what is the new scientific knowledge that this paper has produced.

Third, the paper structure needs substantial changes. The results and discussion sections merged together makes difficult to discern between the actual observations and the authors' analysis. It is essential that the two sections are kept separate. The language used is also not appropriate for a scientific paper: this was mostly informal and colloquial and needs thorough revision.

Fourth, the method was unclear and lacked explanation (at times it was not even easy to understand what cameras have been used, where and how, whilst a schematic would have helped). Overall, this limits the generalisation of the method proposed.

An annotated version is also provided with in-line comments.

*Response: Thank you very much for the comments; we appreciate the time taken to revise our manuscript and the suggestions that contributed to a significant increase in the quality of the paper.*

Line comments:

Line 27: DOI

*Response: replaced*

Line 38: In the field, though. Recently, there has been a lot more work on experimental work to try and define transport dynamics:

Innocenti et al., 2023 https://doi.org/10.1029/2022WR034363

Innocenti et al., 2022 https://doi.org/10.1002/esp.5516

Panici, 2021 https://doi.org/10.1029/2021WR029860

just to cite some that focused almost exclusively on LW transport in flumes

*Response: the reviewer is right; we added more information regarding flume experiments and previous studies; thanks for the suggested references. Lines 32-24.*

Line 56 : typo

*Response: corrected. Lines 65-66.*

Lines 71-76 : This probably needs to be more detailed, as to evaluate what different algorithms do and how they have been adapted to tracking LW and other objects in rivers

*Response: we have extended the section regarding the existing methodolofies for river monitoring using machine learning. Lines 74-89.*

Line 89 : No problem on this, but was there a reason why iPhones were not included? Just because they represent a significant portion of the phone market

*Response: This is something to discuss, but we did not consider iPhones as low-cost mobile phones and we did not have those phones lying around to use.*

Lines 91-92 : Can you add a few more details about this?

*Response: we added more details about the previous studies in the Ain and Allier Rivers in France and the datasets from these previous works. Lines 124-134.*

Lines 102-103 : This is unclear: what are the "rest of the labels"? If 10% is manually labelled, and then the remaining 90% is labelled by means of a CNN, what is the remaining amount of labels?

*Response: We clarified this aspect, that wasn't very clear in the original text. Lines 139-144.*

Line 103 : Would be worth stating the accuracy and how it was checked that

*Response: We added the accuracy. Line 146.*

Line 112 : I noticed that there's a mix of British and American spelling, e.g., 'greyscale' (British), 'labeling' (American). I would recommend to stick to one spelling

*Response: We carefully revised the text and homogenized the style.*

Line 113 : Does it mean this is the total number of LW observed for the whole database?

*Response: We clarified this value and more clearly stated we have 15,228 images with a totalk of 33,160 detection in the database. Line 149.*

Camption figure 2 : Perhaps it may help to have a sub-figure with maps where the images have been captured, rather than just coordinates.

*Response: We added a map as suggested. New Figure 2.*

Line 116 : It is unclear why the PCA is being used here. What is its purpose?

*Response: We explained why this PCA was applied. Lines 149-170.*

Line 125 : This is a rather blunt statement. It can be split like this (and is a fairly common practice), but it is not a necessity.

*Response: We smoothed the sentence. Lines 172.*

Line 141 : By whom?

*Response: We added this missing information. Lines 191-193.*

Line 240 : This needs more detailed explaining what was effectively done: how does the algorithm work?

*Response: we clarified this. Lines 321-330.*

Line 245 : It is difficult to disentangle results from discussion here. Could you not split this into two sections where results are commented separately from any analysis or discussion from the authors?

*Response: We understand the concern, it was challenging to discuss and interpret our results, as this is a methodological paper mostly, and each step and result needed to be justified and explained. However, we improved the structure and split the results from the discussion.*

Line 248 : I wouldn't necessarily call this in scientific terms

*Response: we rephrased it throughout the manuscript.*

Line 250 : This is true for stationary LW, but for waterborne LW?

*Response: we added a clearer explanation. Lines 374-376.*

Line 258 : This needs definition

*Response: we defined this term. Lines 337-339.*

Lines 263-264 : This was already said

*Response: We removed the sentence.*

Line 265 : Surely these are typos?

*Response: Yes, sorry. We don't know how those got through. Corrected. Lines 345.*

Line 271 : This needs to be said in the methods

*Response: we moved this part to the methods as suggested. Lines 379.*

Line 276 : I struggle to follow this section, and why this was needed. Consider re-structuring this paragraph to outline objectives and values displayed in the table

*Response: we restructured the section (Lines 350-354) and moved the explanation to the methods seciton. Lines 308-320.*

Line 285 : Were some cameras attached to bridges? This was not really clear in the methodology. There really needs to be a schematic and an addition to Table 1 with details of the type of camera used (fixed, non-fixed, etc.). Currently it is very hard to understand the setup, as it is quite confusing.

*Response: We elaborated and added some more details in the methods section and to table 1. Section 2.2.1.*

Line 290 : largely smaller?

*Response: we rephrased this sentence. Lines 364.*

Line 295 : Which are...?

*Response: we added more details in the methods. Lines 389-391.*

Line 304 : This is not proper scientific writing

*Response: we changed this term. Lines 398.*

Line 311 : Is this the right word here?

*Response: we removed this term. Lines 415.*

Line 320 : What is 'smaller' here?

*Response: we clarified this aspect. Lines 442.*

Line 339 : How does georectifying come into play here?

*Response: we removed this from here and clarified the sentence. Lines 432-437.*

Line 342 : Such as?

*Response: we added more details. Lines 451-452.*

Lines 342-345 : I don't think this is a proper argument: Neural Networks use statistical relationships that take into account specific characteristics. There are no 'shortcuts' in this

*Response: thank you for the clarification, we rephrased the sentence to make it clearer. Lines 453-459.*

Lines 348-349 : Where online? What was the reason to use this image and not another one? If you found it online, why is it not properly credited?

*Response: We expanded this part of the results and credited the source properly in the methods section. Lines 460-462.*

Line 355 : 'a lot' is very much colloquial

*Response: we replaced this term. Lines 469.*

**RC2 Comments by Dr. Chris Tomsett**

Lines 19-20: There are quite a few more important sources of large wood, such as windthrow and natural mortality, or influence from fauna.

*Response: we have added more sources. Line 18.*

Line 25: I think localised rather larger is more appropriate here when talking about inundation.

*Response: we have adjusted the term larger to localised. Lines 23.*

Line 26: Could likely do with some more up to date references here to show advances in this area. E.g. https://www.mdpi.com/2076-3417/13/18/10454#B65-applsci-13-10454 and https://www.mdpi.com/2077-1312/10/7/911

*Response: we have added one of the suggested references and added another one. Lines 24.*

Line 36: Although this is introductory, I think some reference to those existing algorithms, and why they are location specific would be beneficial, especially as one of the goals of the paper appears to be to reduce the site-specific nature of current wood detection algorithms.

*Response: we have added references to current monitoring sites. Later on in the introduction we talk about why they are site specific. Lines 54-64.*

Line 40: Comma needed after high-resolution aerial surveys. This occurs in other places where you are listing.

*Response: we hadded a comma. And we went through the document adding comma's to listings of 3 or more items.*

Line 41: Give some examples of how RFID is being used, such ashttps://onlinelibrary.wiley.com/doi/full/10.1002/esp.3463?casa_token=G-p1V7DmbDEAAAAA%3AcGP5b3hqKzIPyE8YEHHSPK78ppWrXGMST7iPG-JZUsnuplmrtM2Vs6gkX-LIQYRsjPiCq-bqfgCXEA, https://onlinelibrary.wiley.com/doi/abs/10.1002/esp.1888?casa_token=w1NFWrbZJ7gAAAAA:0YvqxuFyU7vHaDZ2FQ3hHxlDP474jAXCKdoHvR_oKZbKkLphb0btepE7Yw0yjn9ZpJW3KPwQb6tyYQ, and the authors prior work using RFID to improve CNNs.

*Response: we added an additional use of the RFID tags and GPS loggers. Lines 37-42.*

Line 54: Introducing a sentence along the lines of 'they are also limited by their spatial locations, and rely on specific setups being installed prior to an event' This would lead nicely into the use of citizen science.

*Response: we added a similar sentence. Lines 63-64.*

Line 61: Similar to the above, another link sentence here would help the flow, think along the lines of 'Advances in machine learning methods may help to overcome this and allow for widespread wood detection'.

*Response: we have adjusted the last sentence and added a similar sentence to improve the flow. Lines 72.*

Line 62: I am not sure this first sentence is needed, feels informal and unnecessary.

*Response: we deleted the sentence as its contribution is indeed limited.*

Line 67: Starting 'The CNN has multiple…', move this to next paragraph in methods and needs expanding (see comments relating to this below). Then add in some objectives as to how you plan to run, test, and evaluate your algorithm development.

*Response: we have moved this part of the section to the methods section. Lines 92-99.*

Line 78: The first sentence explaining your choice of YOLO algorithm should be proceeded by a small review (one paragraph) on how CNNs work, and why they may be more suitable for detection than other algorithms, building on the information from the introduction.

Following this, I feel that just saying YOLO was chosen for speed and accuracy is limited, especially as this is tested on generic imagery in their paper not large wood. Is there any reason to suggest it would be better for large wood? If not, have other studies trying to detect wood compared between algorithms? Think this is quite a crucial area to justify.

*Response: we have added a section comparing different algorithms and explaining more clearly why we chose the YOLO algorithm. Lines 92-111.*

Line 80: With above changes, a new paragraph could likely be started with 'Training a CNN…'

*Response: we have started a new paragraph there. Lines 99.*

Line 86: Combine these two sentences for better flow.

*Response: we have combined the two sentences. Lines 118-119.*

Line 89-95: I think some more details about the quality of the cameras here would be useful, such as resolution. Do you have a record of how much wood was added to each stream? How long have the monitoring programmes been underway in France and is any of that manual input to the channel? Where are the online videos from and what helps to make the images and wood a more diverse setting? These are all questions that need addressing. Does each image represent a single piece of wood, or are there more pieces of wood in each image, relating to the 15,228 number here. I think you should refer to table 1 here, and also adjust figure 2 as outlined below in figures and tables section.

*Response: we have addressed all questions, clarified these aspects, and referred to table 1. Section 2.2.1.*

Line 98: Figure 2 also shows bounding boxes, maybe reference this instead.

*Response: we have changed the figure we refer to. Lines 138.*

Lines 99-105: This section could do with some additional clarity, especially as this is an additional CCN alogirthm being deployed I assume? Were you checking that the automated bounding boxes for these were detecting wood, as this is not clear which dataset you are referring to by saying the labels were checked manually. There is also no explanation of why this worked better for 11 of the 15 datasets, or what your tolerance for acceptable mAP was, especially as not good enough was below 20 percent. Moreover, when images were checked manually, were incorrectly labelled frames eliminated or adjusted, or left as incorrect?

*Response: We have given a clearer explanation of the labelling process. Section 2.2.2.*

Line 112: Why 80*80, is this purely incidental that no wood was larger than this, I also assume this is in pixel size not other units?

*Response: we have expanded on the explanation. Lines 155.*

Line 116: There is no statement of why this PCA was undertaken, and it only becomes clearer when reading the results. You need to add some context as to why this is undertaken. Furthermore, if the results of the t-SNE test are stochastic, could you not run the test numerous times to assess the diversity, akin to monte-carlo scenarios?

*Response: we have explained why we are using PCA and added an explanation on the application of the t-SNE, which is only used for visualization purposes. Lines 149-170.*

Line 125: Swap must for 'is typically', if smaller datasets don't allow it, there is not always a split in this fashion or a separate test dataset.

*Response: we have replaced 'must' with 'is typically'. Lines 172.*

Line 125 –142: This section is trying to explain a somewhat complex training and validation procedure, whereby computational trade-offs mean omitting some of your data as validation. However, it feels as though how these 6 examples were selected is not overly clear, besides not

being at the same place and time. It may have made sense to use dataset 14 also, purely as that would give you validation samples at a range of sizes.

The section took a while to become clear as to what the process was, and that datasets weren't being dropped from training, just the number of validation sets dropped. Perhaps trying to simplify the wording in places and go through the order. For example, 6 validation cycles were run, for each one a single dataset was dropped for validating and the model trained on the remaining 19. These 6 were chosen to represent a range of conditions, and reduced computational overhead by not undertaking 20 validation cycles.

*Response: we have rephrased most of the paragraph to better explain the training and validation procedure. Section 2.2.4.*

Line 139: Where has this extra dataset come from, and why was it not introduced with the other datasets? Who has been studying this, a research group(s) or monitoring agencies?

This is a useful case example that in essence the paper could have been framed around. I.e. instead of can we implement a cool algorithm, can we reduce human labour of monitoring wood?

*Response: we have added an explanation of the dataset and information on the research in the introduction. Lines 189-193.*

Line 145-152: This seems to be an odd way to do your sensitivity test, as although you are trying to identify the effect of number of inputs on output quality, if these inputs are multiplied for smaller datasets, then they are not adding any extra information, only overtraining the model? Would it not have been better to undertake this at a smaller number of images to assess performance, on possibly a limited number of datasets. I.e. for 10 of your datasets over 300, or 8 over the 1000, sequentially go from including 100, 200, 300, etc, and then quantify at what point there is no improvement in the model? This in itself could be one of the scenarios.

*Response: We agree that the suggested way would also have been an excellent sensitivity analysis. We did it in the described way because we wanted to keep most of the data in the large datasets, without overrewarding the large datasets. This is because the model is biased towards large datasets as it is rewarded equally on each image on the total database. We have added this explanation to the text. We made a separation between first testing how much data was needed in the training process before testing how to improve the performance of the model in the next section. We have explained this in a better way. Section 2.3.1.*

Line 154: Although it is clear there are no river based large wood studies to learn from, it seems that casting the net a little wider shows these studies have been used in similar ways on living trees and perhaps other wood related scenarios. E.g. https://ieeexplore.ieee.org/abstract/document/9643113? casa_token=Vm749u_aLtQAAAAA:IQ8hGqEscqOOTf4M5Co8uVAJ1qsiJtDGUoMrQDFj- oSM14tTKiVBbKzIUl1G0OTwZ5AGgRy_qw

*Response: we clarified this and added that a CNN had not yet been trained for our specific purpose.* And we added a reference to the referred article in the introduction. Lines 76-89.

Line 158: Although in principle I can understand how all these parameters effect wood detection, but has any worked actually been conducted on this? If so, reference it.

*Response: we have added an explanation regarding augmentation strategies transferring poorly between datasets, and that it's why we explored different strategies. We have added a reference on how to augment datasets, but we haven't found a source that specifically explores training scenarios for wood detection purposes. Lines 209-220.*

Lines 158-160: It says 14 were trained and compared to a baseline, but there are 13 outlined. Either say 13 models trained, or 14 including the baseline for which the other 13 are compared to.

*Response: we have changed it accordingly to 14 including the baseline. Line 209.*

Lines 162-163: Why were the values of 4% and 30% chosen, is there a rationale for this? The logic behind this makes sense, but just need to clarify reasoning for thresholds, even if they were just decided as no previous study to base upon.

*Response: we have added more explanation on how we got those numbers. Lines 221-228.*

Line 170: Can the dataset size be given for total number of images, i.e. how similar is 'approximately the same'. This also feeds into informal language comments.

*Response: we have added the dataset size. Line 234.*

Line 171: Why such a high number, when lowering it slightly could reduce the need for oversampling from some datasets? This appears similar to the sensitivity analysis you performed.

*Response: we have clarified why we chose this number. Lines 236-237.*

Lines 176-179: Can you specify the number that were rotated vs mirrored, as the and/or makes it unclear if this was randomly done and randomly distributed. Were any both mirrored and rotated?

I agree, that only partial rotation is necessary, this seems like a sensible decision to have made.

*Response: we have explained the total number of mirroring and rotations performed. Lines 241-247*

Line 180: Again, how many were altered, and what proportion were mirrored or rotated. I think this needs more detail so the user knows what was done. How much extra data did this result in?

*Response: we have added an explanation of the total number of images in the dataset for this scenario, and also for the two scenarios after. Lines 248-251.*

Line 188: I feel that the inclusion of the phrase 'non–living wood' implies you are adding living wood, as opposed to wood that is not floating. Could be removed.

*Response: we meant that the there may be a difference between detecting living trees and wood, but we understand the confusion and removed it. Lines 258-262.*

Line 189: Change example to wood sample.

*Response: we have changed example to wood sample. Lines 260.*

Lines 192 –196: This is an interesting scenario, primarily as these are open source datasets, with lower quality, but greater geographical diversion. In essence, I am not sure this is just testing data quality. Again, with the addition of other datasets, I think they should be mentioned in the original introduction of data, and their locations (even if approximate) included on a figure map. They can be highlighted/commented that they are only used for testing or specific scenarios, but curious as to why they were not included from the outset?

*Response: We agree that we are also testing the geographic diversity of the data in this scenario. We did not include this data because we labelled the data only in a later stage. The tests before were already performed by that time. Therefore, this is not part of the main dataset and is merely a test to check whether adding any data of even low quality would help in training the model. As this part of the test is small, we do not feel like further elaboration on the data is justified.*

*Regarding data quality, YouTube and Twitter generally highly compress videos, and therefore, the data quality is generally lower. We have added this aspect to the discussion. Lines 263-269.*

Line 199: Which datasets were removed?

*Response: we have added the datasets that were removed, and clarified this aspect in the revised text. Lines 274.*

Lines 201 –206: This is really well explained and justified here, so should therefore be a model for your other scenarios where the justification is weaker.

*Response: we have added more detailed explanations to the other scenarios. Lines 276-281.*

Lines 207-211: This is an interesting scenario to assess, as many secondary data sources may be of lower quality. However, are the double- precision images used either a) the down sampled images at 416*416 resampled again to a higher resolution, or b) the original images resampled to 832*832? The text make it seems like you double the resolution of the down sampled image (scenario a), as opposed to changing the original resampling (scenario b). Make this clear either way.

*Response: we have changed the explanation and made clear it is scenario b. Lines 282-286.*

Lines 213-226: This is a really well written section on the statistics being used, what they mean, and how a reader should interpret them.

*Response: thank you.*

Lines 232–239: How is this sensitivity test different to the one introduced earlier, and why has this one got more dataset sizes to test the sensitivity? This is also not referred to in the results as far as I can tell, so what is the purpose of this section?

*Response: This is indeed not different and we have removed it from the manuscript.*

The variance method also adds some confusion, is each of these models run several times, and the best results taken? If so, why the best results, does that not overestimate model performance? This could be explained better.

*Response: We have elaborated on the explanation of the process in lines 310-320.*

You then talk about comparing between models, which again is fine but is very brief as to why, needs more explanation. You then mention a final model, is this not just your optimal model from all your testing?

*Response: We have changed the section to explain why our method actually stops the overestimation of the models' performance. Lines 311-313.*

Line 238: Which dataset is this, is it the same as the one introduced previously on the river Inn? Again, this needs to be stated.

*Response: we have elaborated on the Inn dataset further in section 2.1.3 and in this part referred to that section.*

Lines 240–244: This seems like a really sensible addition and is good to see some unpicking of what is happening behind the scenes. Maybe a brief idea of how this works, and what you hope to find and why you picked certain images (one river, across rivers, different angles?) would be sensible? In this case you might hope to hypothesise why some images/datasets are less well classified? This would be nice to see expanded on in the discussion.

*Response: We have extended this part to better explain what we are trying to understand from the method. Lines 321-330.*

Line 248: I am not sure 'blob' is appropriate here, and if they are so small how can you be certain these are pieces of wood? You mention wood remaining stationary, does that mean moving wood was not included in the study?

*Response: We have moved this part to the discussion section, and adjusted the terminology throughout the manuscript. Also, we removed the word stationary as it was confusing.*

Lines 254–264: Unfortunately, no supplementary could be found on the online interface for comparison. However, I do wonder if whether double panelling a figure to include one of these plots for clustering with figure 5 could help to show the variation.

I would also argue that the relative sizes of the bounding boxes compared to images were not that different, with many similar distributions and a few outliers, primarily from external datasets which is to be expected.

You also state that for 12, 18, and 19, the drop in relative size could be due to low camera resolution or distance from stream, but 12 is one of the model setup cameras so surely you know this, and could tell for the others by looking at the original images?

*Response: we have added supplementary material, and included a figure to the revised manuscript. Also, we have added a part in which we compare images from different datasets in the*

*supplementary material and explain what we mean by different in quality. In fact, including some of this data posed additional challenges, as it was hard to identify and label the wood even manually.*

Line 265: Assume this is meant to be Database Configuration.

*Response: Yes, correct. Something went wrong here. Lines 345.*

Lines 266–270: As this is both a results and discussion section currently, there is a lack of discussion here about why this may be, and that by oversampling images you may not see an improvement in model performance purely due to the model become more tuned to those specific examples.

*Response: we have decoupled the two sections, expanded the results part and added a separate part in the discussion. Lines 379-383.*

Line 270: This is very important, if you do not now oversample, in your scenarios where you mentioned oversampling smaller datasets, did you now not do this? This seems like quite a big change. If so, I think the sensitivity results need to come within the methods inclusive so that you do not explain changes in your methods during the results.

*Response: The text was unclear. We have not used these results to adjust the methods, and now we clarified this and deleted.*

Line 273: What were these results, and are they really comparable considering the differences in the object types?

*Response: The results have a comparable mean average precision. We have added this to the section. Lines 377.*

Lines 274-276: This section is not overly clear, I think it needs better wording to explain what is being done here, especially regarding the multiple training rounds. This feeds back into above comments at the end of the methods.

*Response: This line summarized a larger part of the methods section in 1 sentence, and was indeed unclear. Therefore, we removed this sentence and indicated that the table shows the results from the training scenarios as explained it in the methods section. Lines 310-320.*

Lines 281 –284: I can see what is trying to be said here, about training for specific or general wood detection, but feel it could have been said better. This is also the first mention of how cameras were mounted, perhaps this should be mentioned in the data section also.

*Response: we have added information on the mounting points of the cameras in the methods section. Also, we have adjusted the explanation to be more clear. Lines 355-359.*

Lines 285 -293: There is a focus here on the high–definition wood images in this analysis, and yet there are only 9 images in the dataset. As such, are larger changes in mAP not more likely due simply to the lower number of objects to compare against? This is somewhat shown by the weighted average, and so overstating the importance of a vast performance decrease or increase here may be unjustified. The narrative however, that good wood images lead to

better training than poor wood images, is justified by the average and weighted average outputs.

*Response: It is correct that if there are fewer samples in a validation dataset, the model missing 1 more piece of wood already drastically decreases its performance. However, all samples in the dataset are clearly pieces of wood with explicit characteristics of wood. So the model missing one or two more pieces does indicate that is it not as good at understanding the characteristics of wood. That being said, the fact that we state 'vastly' and show a large percentage is indeed unfair. Therefore we have revomed the statements of the amount of decrease. Lines 360-368.*

Lines 294– 295: Has a significance test been undertaken here?

Are these broadly speaking not the only two factors, apart from manual labelling to begin with for training.

What are the worst performing models?

*Response: we have now explicitly mentioned which models performed best and worst. Also, as the data quality seems to be the larger limiting factor, we have removed the algorithm from the statement, as it performs better with different datasets. Also, we changed the word significant. Lines 386-389.*

Lines 296 –297: This sounds like you have added in an extra scenario, rather than describing one of your scenarios.

Change 'where the datasets with lower performance than 30% mAP were excluded' to 'where the datasets with a mAP of lower than 30% were excluded'.

*Response: we are talking about scenario 11 here. We have added that information to the section. We also changed the wording following the reviewer's advice. Lines 390.*

Lines 300– 301: Which scenario is this, can't find a reference to 19% in the table that is positive? If this is just assuming the inverse, then the addition of these images back wouldn't be the same 19% as the base conditions would be a different value.

*Response: we are trying to describe the opposite, so the -19 percentage points we see with scenario 11 can be interpreted as a +19 percentage points when we add data of the same scene but from another day. We have made this explanation clearer in the text. Lines 392-396.*

Lines 301 –306: This is a really important and useful point, and should be one of the key take home messages that adding to existing databases with some data from a site improves the algorithms performance. Check some wording here though, especially when speculating performance benefits.

*Response: we altered this part. Lines 394-401.*

Lines 307–313: This is an interesting section about whether the time component is critical. However, I fell it is overplayed in its significance. Of the two worst performing datasets (11 and 18) only one shows an increase of 6%, the other a decrease. Therefore, to say improvements of nearly 10% are made is an

exaggeration. Arguably, this is somewhat upstaged by the large decrease in one of the better performing datasets (3).

*Response: we agree, and have rephrased it, although we believe it is still very interesting for future research. Lines 411-417.*

Lines 313 –316: Make this a separate paragraph as it feels separate from the temporal component.

Compared to the emphasis placed on scenario 12, scenario 13 appears to show much greater performance gains, and the importance of image resolution in tracking wood. As this has implications for how wood should be monitored, both from a hardware and software perspective, it likely needs more attention and discussion around the trade-offs between image resolution, computational efficiency, and expected wood size.

*Response: We have elaborated the disussion on scenario 13 in its own paragraph. Lines 418-423.*

Line 315 references image 5, is this from figure 2 as these seem to be larger wood size, if not, please be clearer as to what this refers to.

*Response: we meant to say that from the 6 reference datasets, the 3 that show the greatest improvement are the one that have the smallest relative Bbox sizes. We have made this more clear in the text. We have also elaborated on the implications for practitioners. Lines 421-423.*

Lines 317 – Onwards: This almost feels like a different section or subsection, as it is a change from training and validating to assessing the model used. It seems as though this section itself however is limited in just comparing two models, moreover, these results have differences greater than many of the scenarios provided above, which indicates that model choice may be more important than datasets, something that is not discussed in great detail. As a result, the take home would switch from the importance of data, to the importance of model selection in getting the best outputs…

*Response: we have created a new section for this part. The differences between the models are indeed larger than between many of the scenarios. We have adjusted the text as well. Section 4.2.*

Line 329: Perhaps, if a new subsection is introduced for the above, this should be moved prior to it.

*Response: we have moved this part up. Lines 426-437.*

Line 334: Reference figure 7 here, as it is not referenced anywhere in the text

*Response: we have referred to the methods section where we have elaborated on this dataset. And we have referred to figure 7. Lines 425.*

Lines 335 –337: You identify that the model is better at identifying large wood, and then state how large wood components compromise the greatest proportion of transport, but this needs to be referenced to support this. Furthermore, small wood components also play a role in increasing the total volume of log jams etc and so important to monitor. Commenting on how this is missed in the dataset is probably needed.

If possible, it would be great to look at those that are missed and estimate the size of these to identify a limit of detection. However, that may be beyond the scope of this investigation and potential for future research.

*Response: We have added references. Lines 432. It would indeed be interesting to find that out. And we are working on that, but is indeed beyond the scope of this manuscript.*

*We have adjusted the section to also indicate the importance of small wood and measured that can be taken to have a better change of detecting small wood also.*

Line 338: Have these images been georectified in the processing? If so this needs to be explained for reproducibility. Moreover, if they have then they could be used to identify the limits of detection for wood as per above?

*Response: no, they have not been georectified in this study. We are elaborating on the potential of the method. However, as this was confusingly phrased, we changed the wording. Lines 432-437.*

Line 342: Give examples here please, and comment on how they may differ or align to wood detection (e.g. shape and background).

*Response: we have added an example (humans walking through the frame). Lines 455-459.*

Line 342–346: think this needs to be reworded, at times this sounds speculative and also non-scientific. The theory of not being able to detect outside of the training sample is sound, just the transmission of this information is not clear enough.

*Response: we have adjusted this section and added examples. Lines 451-452.*

Line 347: Where was this from and why not use one of your current data? Again, this points to questions going back to your initial data introduction, and consistently adding new bits of information.

*Response: we have elaborated on the source of the image and added it to the methods section and better explained why this image was used. Lines 460-462.*

Lines 350– 360: Does this not come back to simple survey and image design. If most of your images are from roads and bridges overlooking rivers, and you provide an image much closer to the channel, it will struggle, until as you say you include images of large bits of wood close up. Therefore, to use the word remarkably again seems a little overstated.

*Response: with the word remarkable we do not necessarily mean something positive. More like something we might not have expected. Lines 464-475.*

Lines 357-358: Can you expand on how you know it is using the wood texture, is this hypothesised from the location of the pixels used, or can this be proven?

*Response: you are right, we have no way of knowing that the model actually understands the texture of wood, and therefore the wording was too enthusiastic. We have changed the wording to be more careful. Line 472.*

Lines 363–371: This is a nice start to your conclusion, summarising your results well to give an overview of the paper. However, there is no comment on how increasing data sizes or changing their angles/mirroring had no effect.

*Response: we have added this to the conclusion. Line 493.*

Lines 372 –382: I feel that to say your model struggles on the definition of wood, unless its given high-quality images of wood not in rivers, is overly harsh on your model. The purpose of this paper and method is to detect wood in rivers, likely from monitoring stations above the rivers surface (on bridges etc). So the model works if it detects these well, and shouldn't necessarily be able to detect wood such as in Figure 8. Therefore, the model CAN generalise the concept of wood 'in rivers', which is the main purpose is it not?

I think the word blob should be removed throughout, perhaps in this instance they are best referred to as fragments or segments, i.e. not all the wood is on show? Make sure this distinction is first explained when replacing the initial occurrence of the word 'blob'.

This may be clarified by an earlier point, is this 19% increase simply the opposite of the 19% reduction when the Allier dataset (18) is removed? If so. This is not 19% (e.g. 20% decrease from 100 is 80, a 20% increase from 80 is not 100). If this is a separate analysis, make sure this is clear during the methods and results. It could even be viewed as an additional scenario (e.g. adding same site from different date).

*Response: We have adjusted the explanation about the model understanding the concept of wood.*

*-we have removed the word blob from the paper.*

*-you are right that we made a mistake in explaining the percentages. We have adjusted the explanations to the words 'percentage points' where we made this mistake.*

Lines 383– 387: This could likely be grouped into areas of future research. 1) real time monitoring 2) algorithm development and miniaturisation 3) temporal imagery for object detection. These could also form some structure for a separated discussion, allowing room to discuss the impacts of the research.

*Response: we have adjusted the last part to more clearly touch upon those points. Lines 507-511.*

Figure 1: This figure could benefit from labelling the boxes with the sections of the method that they refer to. This will allow readers to quickly understand which bit of the process they are referring to. Make sure the naming matches to, it will help the reader.

This could also be improved by creating this as an overall schematic of the methods, which would better describe the whole process as mentioned prior.

*Response: we have adjusted the titles in the text and in the figure to correspond. We have also added the section numbers to the figure.*

Figure 2: It is great to see some visual examples of what these images look like, and how they differ, especially in regard to the additional imagery. However, I think it would be good to possibly

remove one or two images, and add an inset location map showing where in the world these were taken from, rather than coordinates in the caption. This would give a better idea to the readers of where your data is coming from. You could colour or size location dots based on the number of images from a location as well.

*Response: we have added the locations to the images instead of coordinates.*

Table 1: Could this table also have a column or some stars which denote the datasets used in validation, these are mentioned later on but will help the reader when scanning back and forth. Consider making either camera lowercase, or the unknown and differing upper case.

*Response: we have indicated the 6 representatative datasets and added more information about the cameras.*

Figure 3: Why is this figure not further up in the manuscript? It is referenced first several pages earlier and causes confusion in the current section. Appreciate this may just be a current formatting error for the preprint.

*Response: it is now further up in the manuscript.*

Figure 4: No changes required for this figure, it is clearly laid out, shows the size of datasets, and helps to explain what is happening in terms of the number of training vs validation datasets.

*Response: thank you, we kept it as is.*

Figure 5: Again, another clear figure which adds to the manuscript and is broadly easy to interpret. The inclusion of a double headed arrow along the x axis, pointing to larger wood and smaller wood may help with interpretation, so readers know if the value is indicating a lot of the image is the woods bounding box, or little.

*Response: thank you, we kept it as is.*

Figure 6: This figure is good, however it could do with stretching along the x axis, as this will help to show the variation in IOU training loss which show subtle differences.

*Response: thank you for the suggestion, however the detail in the image does not allow for stretching. The bandwidth between the epochs is already clear in our eyes.*

Table 2: The table layout is fine, but the text is a little hard to read in places. For those reading in non-colour or with colour-confusion, perhaps as well as colours a marker could be used to quickly attribute greater than 3% increases or decreases.

*Response: colours are actually not allowed in esurf tables, so we indicated them with stars.*

Figure 7: A useful figure, make sure it is referenced in the text. Are these bounding boxes ones predicted by the model or drawn manually for users. It could be better to include boxes created by the model as well to show the types of

wood it is missing (perhaps detected and missed wood as two separate colours?).

*Response: we want to show an example of the dataset here and not qualitatively go into specific pieces that are missed by the model. For the analysis, we only use mean Average Precision in the text. We have referenced it in the manuscript.*

Figure 8: Are the bounding boxes in this figure manually drawn? If so, they should probably better align with the extent of the wood. Likewise, as the percentage is referring to overlap in bounding box size, perhaps indicating the bounding box of the detected wood would help to illustrate these differences? Otherwise, this is a very helpful and useful figure.

*Response: The bounding boxes are created by the model and are, therefore, not perfect. Here we go into the detections qualitatively. We have adjusted the explanation of the image to stress these are model generated boxes.*

---

## Referee Report (RR1)

Thanks for the revised manuscript, which contains a much improved version of your manuscript, and that addressed most of the major points raised previously.

Regarding my previous major points, I note the authors effort, and specifically:

**1.** I understand the authors added more information in the introduction about different CNNs and other deep learning techniques. I think what is still missing is in the discussion section a paragraph where the authors discuss what other neural networks could (or could have not) achieved, based on the existing literature. This will help to underpin the authors' method as the "way to go" for automatic detection.

**2.** It is now much clearer what the aim of this paper is from the onset.

**3.** The paper structure is now much more readable and I appreciate the authors' effort to achieve that.

**4.** Whilst the methodology is substantially improved, some aspects are still a bit unclear to me, specifically about where and how the cameras were placed. I think a schematic would really help (even as a subfigure to figure 2 or a standalone figure), as also suggested in my previous review. Also, for Figure 2 the map is very welcome; however, the locations are impossible to identify, I recommend in-set maps to show where bridges/capturing points are located.

**Other minor line-by-line comments:**

General: I note the effort done by the authors to use one type of spelling only. There are a few still to be solved (e.g., utilized, analyze, hypothesized, greyscale now turned into grayscale)

Line 83: "which we attribute to the lack of uniform data" is there a reference(s) that you can use to substantiate this?

I note the response from the authors about iPhones (and by extent to many other brands) which is perfectly fine. However, can they add the information provided in the text (i.e., that the phones used in the study were those only available to the authors, so there could be uncertainty about other brands and processing software)?

Line 92: CNN and YOLO have already been defined above, so no need to re-instate the acronym

Line 150: "a lot of", this is not formal scientific writing

Line 150-151: "Naturally, the wood floats and moves in a f low or is deposited or trapped by an obstacle (i.e., river bank, boulders, trees), some videos contain minutes" sentence needs rephrasing

The authors mention that they credited the images used for the dataset, but this is not reported in any of the figures or in the text, please correct.

Line 638: whilst I thoroughly appreciate the authors' encouragement for this reviewer to get a promotion to full professorship, I am still a "Dr"

References: please note the typo in "Panici (2021)" and in-text citation

---

## Referee Report (RR2)

**Overview**

Thanks to the authors for taking on-board the suggestions of the prior reviews and clearly outlining the changes they have made.

The modifications to the paper have helped to improve its overall quality and robustness. Special attention has been paid to make the methodology clearer to the reader and as such it is easier to follow and understand what has been undertaken. Likewise, inclusion of more detail about how changes to data were undertaken and the number of images and samples has helped to make it clearer the volumes of data used overall and in the different scenarios.

There are a few outstanding changes which are advised below, and with this the work is of publishable quality and will be of benefit for researchers in this field in the future.

**General Comments**

| Section | Comment |
|---|---|
| Introduction | The introduction is improved, both by clearly stating what the paper is aiming to do and giving a broader overview of the current literature on the topic. Likewise, the shift in introducing CNNs to the method section helps to create a better break point in the front end of the paper. |
| Methods | The choice of the YOLO algorithm as a starting point here is much better justified, alongside providing context to other algorithms that could be used and why they were not. This is a considerable improvement over the prior version of the paper.

The sources of data are also much better explained, and the variation in capturing technique and methods (natural vs seeded). This is also echoed in the data description for table 1. Likewise, the PCA purpose is much better explained and clearer for the reader which is excellent to see.

Although section 2.2.1 may be better using a range of values, it is now better justified as to finding whether less data would decrease performance or not as opposed to finding an optimal size. However I think as you have some different sized scenarios, and the 2000 sensitivity example will be using multiples of the same image, I feel this section could just be removed with no impact on the paper.

The inclusion of all the datasets in the scenarios really helps to see how much data is being used in each. Changes to the following sections also help to distinguish between what the scenarios are doing and any following tests that were undertaken. |
| Results | The section on training data clustering is much improved, with the changes made helping to inform the reader of the variation and/or similarities in your data.

Again, for the sensitivity, I think unless you could rerun to look at say 250, 100 images as well this isn't rally adding much to the narrative or useful to |

| | |
|---|---|
| | the reader as it hasn't identified a threshold at which time and performance separate/converge.

I feel that with the separation of the results and discussion, although this has helped distinguish between what you have found and why that may be, it would make more sense still for the test on the river Inn to be placed in the results. This should be relatively straightforward with a split halfway through that paragraph in the discussion. I see less of an issue of the YOLO algorithm results being in the discussion, as this is more or a discussion of the algorithm selection than an aim of the paper to find results of.

Overall though the separation of the two sections has been done well and is easy to read and follow. |
| Discussion | Clarity around the discussion relating to adding similar images was welcomed, especially from the same river at a different time point. There were some clarifications in calculations here too. The addition of discussions around earlier scenario tests was also welcomed as this was missing prior and provides useful guidance for database configuration. However, there still seems to be no reference to the augmented scenarios around rotation and mirroring (apart from a line in the conclusions). These appear to show changes in line with other scenarios but are not overly discussed. Adding a sentence or two discussing these after the sampling discussion would be sensible, even if to mention the impact not being as much as expected.

There may need to be some reference to the visual similarity of images in for the river Inn experiment, was this likely to be a good match to the training images for example?

Overall though the discussion is well framed in which each component being discussed is countered with a point about how this may impact future practitioners in the field which is useful for the paper to have both applicability and need.

I wonder if for using the Yolime on images, whether the same could easily be replicated for a standard image from one of your bridge datasets perhaps? You note the importance of surrounding water to detect wood, perhaps this would also be shown in your other images? This could then be combined with your current figure 8 to have 4 images, two algorithsm at two locations?

It was good to see the addition of low-cost camera limitations, but perhaps adding a comment weighting up the benefits vs drawbacks of low-cost cameras for science vs monitoring could be included. I.e. are they good enough for monitoring and alerting of large wood flows, but not good enough for quantifying and understanding wood transport dynamics? |
| Conclusion | The conclusion has been reworded in to succinctly summarise the substantial amount of work that has gone into this paper. This is effective at displaying the key messages to the reader. |

**Specific Comments**

| Section | Line | Comment |
|---|---|---|
| Introduction | 83 | Start a new paragraph with 'in this study', helps to really showcase what the aim is rather than being lost in the last introduction paragraph. |
| Methods | 137 | Can you add 'both using manual and automated approaches' to the end of the first sentence (or similar), just so that readers are aware from the outset that there is a mix and that's why the pseudo-labelling is introduced. |
| | 153 | Was this a specified percentage or did it vary, needed to be included to decide what counts as overlapping? |
| | 189 | Although this section is now much clearer with the introduction of an extra validation dataset. I still believe including this in the data acquisition section would be best to avoid confusion over bringing extra datasets in. Alternatively, adding a statement in the data paragraph that additional data is used for validation or in some scenario cases would also be sufficient. |
| | 214 | Add in 'for' between points and mean. |
| | 269 | The sentence here relating to the figure does not make sense. A) I think you mean figure 5, and B), change an 11 and A 12 to 'The 11 and the 12 descriptors are the self-gathered extra datasets'. |
| | 274 | Why was dataset 18 not removed as well if it is shown to be one of the worse? |
| | 286 | Just to clarify, as I don't think this is particularly explicit still. Are these images resized to 832x832 from the original image, or by 'doubling' the resolution of the sampled 416x416 images. Mainly as doubling the resampled images implies something about the image size (in bytes), whereas resampling to 832 tells you something about image detail retained. |
| | 316 | New line at 'After that' and perhaps change this to subsequently or additionally instead. |

| Results | 352 | Insert 'test' or 'database configuration' before scenarios to distinguish from baseline scenario. |
|---|---|---|
| Conclusion | 507 | Change can not to cannot, although correct 'can not' over emphasises the negative, whereas in reality there is potential for real-time use. |
| | 509 | Merge the two sentences into one here, there seems to be no benefit of separating them out. |

**Figures**

| Figure | Comment |
|---|---|
| Figure 2 | Could this include a reference to the source? Whether its primary collection, secondary (i.e. the Allier and Ain), and presumably purchased imagery for number 9? Either in the figure or the caption. |
| Table 1 | The inclusion of the number of unique labels is great for working out which sites may be contributing more to influence the model performance. |

---

## Author Response (AR2)

Thank you for the comments. In the following, we reply to each of your comments.

1. I understand the authors added more information in the introduction about different CNNs and other deep learning techniques. I think what is still missing is in the discussion section a paragraph where the authors discuss what other neural networks could (or could have not) achieved, based on the existing literature. This will help to underpin the authors' method as the "way to go" for automatic detection.

- We have added a subsection (4.4) that addresses the mentioned discussion.

2. It is now much clearer what the aim of this paper is from the onset.

- Thank you, we agree.

3. The paper structure is now much more readable and I appreciate the authors' effort to achieve that.

- We agree, thanks to you as well.

4. Whilst the methodology is substantially improved, some aspects are still a bit unclear to me, specifically about where and how the cameras were placed. I think a schematic would really help (even as a subfigure to figure 2 or a standalone figure), as also suggested in my previous review. Also, for Figure 2 the map is very welcome; however, the locations are impossible to identify, I recommend in-set maps to show where bridges/capturing points are located.

- We have added the coordinates to the caption of the figure, so the people interested can find the exact location. We have also added a more elaborate description of how we mounted the cameras (lines 123-125), and have added an example to the supplementary material.

Other minor line-by-line comments:

General: I note the effort done by the authors to use one type of spelling only. There are a few still to be solved (e.g., utilized, analyze, hypothesized, greyscale now turned into grayscale)

- We have gone through the text and adjusted.

Line 83: "which we attribute to the lack of uniform data" is there a reference(s) that you can use to substantiate this?

- We have added references.

I note the response from the authors about iPhones (and by extent to many other brands) which is perfectly fine. However, can they add the information provided in the text (i.e., that the phones used in the study were those only available to the authors, so there could be uncertainty about other brands and processing software)?

- We have added that indeed these phones were the phones available to the authors.

Line 92: CNN and YOLO have already been defined above, so no need to re-instate the acronym

- We have deleted the re-instatements of the acronyms.

Line 150: "a lot of", this is not formal scientific writing

- We have remove the term 'a lot of'.

Line 150-151: "Naturally, the wood floats and moves in a f low or is deposited or trapped by an obstacle (i.e., river bank, boulders, trees), some videos contain minutes" sentence needs rephrasing

- We have rephrased the sentence to be more clear.

The authors mention that they credited the images used for the dataset, but this is not reported in any of the figures or in the text, please correct.

- Where we were not the data acquirer/creator, we have added the credits.

Line 638: whilst I thoroughly appreciate the authors' encouragement for this reviewer to get a promotion to full professorship, I am still a "Dr"

- We have changed the title, next time we might have to write 'Prof.'.

References: please note the typo in "Panici (2021)" and in-text citation

- We have corrected it, honest mistake.

---

## Author Response (AR3)

Reaction to the comments of Chris Tomsett (14 October 2024):

**General comments:**

Introduction
The introduction is improved, both by clearly stating what the paper is aiming to do and giving a broader overview of the current literature on the topic. Likewise, the shift in introducing CNNs to the method section helps to create a better break point in the front end of the paper.

- We agree with this. It is improved drastically.

Methods
The choice of the YOLO algorithm as a starting point here is much better justified, alongside providing context to other algorithms that could be used and why they were not. This is a considerable improvement over the prior version of the paper.

The sources of data are also much better explained, and the variation in capturing technique and methods (natural vs seeded). This is also echoed in the data description for table 1. Likewise, the PCA purpose is much better explained and clearer for the reader which is excellent to see.

Although section 2.2.1 may be better using a range of values, it is now better justified as to finding whether less data would decrease performance or not as opposed to finding an optimal size. However I think as you have some different sized scenarios, and the 2000 sensitivity example will be using multiples of the same image, I feel this section could just be removed with no impact on the paper.

- Even tough we agree that there is overlap between the scenario's and this initial analysis, we still think that the purpose of this analysis is slightly different and would therefore like to keep it in the manuscript. The scenario's are more looking at the composition of the datasets whilst this initial analysis only looks at the quantity.

The inclusion of all the datasets in the scenarios really helps to see how much data is being used in each. Changes to the following sections also help to distinguish between what the scenarios are doing and any following tests that were undertaken.

Results
The section on training data clustering is much improved, with the changes made helping to inform the reader of the variation and/or similarities in your data.

Again, for the sensitivity, I think unless you could rerun to look at say 250, 100 images as well this isn't rally adding much to the narrative or useful to the reader as it hasn't identified a threshold at which time and performance separate/converge.

- We agree that further analysis would be beneficial, these are the analyses we did and can still save others work by not having to label 2000 images per dataset.

I feel that with the separation of the results and discussion, although this has helped distinguish between what you have found and why that may be, it would make more sense still for the test on the river Inn to be placed in the results. This should be relatively straightforward with a split halfway through that paragraph in the discussion. I see less of an issue of the YOLO algorithm

results being in the discussion, as this is more or a discussion of the algorithm selection than an aim of the paper to find results of.

- We agree and have separated the River Inn performance and added a \subsection{Test results} to the results.

Overall though the separation of the two sections has been done well and is easy to read and follow.

Discussion
Clarity around the discussion relating to adding similar images was welcomed, especially from the same river at a different time point. There were some clarifications in calculations here too. The addition of discussions around earlier scenario tests was also welcomed as this was missing prior and provides useful guidance for database configuration. However, there still seems to be no reference to the augmented scenarios around rotation and mirroring (apart from a line in the conclusions). These appear to show changes in line with other scenarios but are not overly discussed. Adding a sentence or two discussing these after the sampling discussion would be sensible, even if to mention the impact not being as much as expected.

- We have added a reference to the augmentation scenario's to the discussion.

There may need to be some reference to the visual similarity of images in for the river Inn experiment, was this likely to be a good match to the training images for example?

- Like the other datasets, the river is contained of images that were taken by a camera that was attached to a bridge. However, the size of the river was way larger at the Inn River and therefore actually was quite different from the training data. We have added this to the discussion.

Overall though the discussion is well framed in which each component being discussed is countered with a point about how this may impact future practitioners in the field which is useful for the paper to have both applicability and need.

I wonder if for using the Yolime on images, whether the same could easily be replicated for a standard image from one of your bridge datasets perhaps? You note the importance of surrounding water to detect wood, perhaps this would also be shown in your other images? This could then be combined with your current figure 8 to have 4 images, two algorithsm at two locations?

- Although we agree that this can be an interesting analysis, we decided to leave it as is and keep it for further investigation.

It was good to see the addition of low-cost camera limitations, but perhaps adding a comment weighting up the benefits vs drawbacks of low-cost cameras for science vs monitoring could be included. I.e. are they good enough for monitoring and alerting of large wood flows, but not good enough for quantifying and understanding wood transport dynamics?

- We have added a comment weighting the benefits also.

Conclusion
The conclusion has been reworded in to succinctly summarise the substantial amount of work that has gone into this paper. This is effective at displaying the key messages to the reader.

-Thank you.

**Specific comments:**

Start a new paragraph with 'in this study', helps to really showcase what the aim is rather than being lost in the last introduction paragraph.

- Done

Can you add 'both using manual and automated approaches' to the end of the first sentence (or similar), just so that readers are aware from the outset that there is a mix and that's why the pseudo-labelling is introduced.

- Done

Was this a specified percentage or did it vary, needed to be included to decide what counts as overlapping?

- Added more detailed information.

Although this section is now much clearer with the introduction of an extra validation dataset. I still believe including this in the data acquisition section would be best to avoid confusion over bringing extra datasets in. Alternatively, adding a statement in the data paragraph that additional data is used for validation or in some scenario cases would also be sufficient.

- Moved to data acquisition.

Add in 'for' between points and mean.

- Done

The sentence here relating to the figure does not make sense. A) I think you mean figure 5, and B), change an 11 and A 12 to 'The 11 and the 12 descriptors are the self-gathered extra datasets'.

-A Correct, the referred figure was incorrect, adjusted.
-B Done

Why was dataset 18 not removed as well if it is shown to be one of the worse?

- We explained why we removed 12 and 19 and not 18.

Just to clarify, as I don't think this is particularly explicit still. Are these images resized to 832x832 from the original image, or by 'doubling' the resolution of the sampled 416x416 images. Mainly as doubling the resampled images implies something about the image size (in bytes), whereas resampling to 832 tells you something about image detail retained.

- We added a more clear explanation from which resolution the original images were resized to.

New line at 'After that' and perhaps change this to subsequently or additionally instead.

- Done

Insert 'test' or 'database configuration' before scenarios to distinguish from baseline scenario.

- Done

Change can not to cannot, although correct 'can not' over emphasises the negative, whereas in reality there is potential for real-time use.

- Changed. The word 'yet' (as in cannot yet) shows that there is potential.

Merge the two sentences into one here, there seems to be no benefit of separating them out.

- Done

Figure 2: Could this include a reference to the source? Whether its primary collection, secondary (i.e. the Allier and Ain), and presumably purchased imagery for number 9? Either in the figure or the caption.

- Done

Table 1: The inclusion of the number of unique labels is great for working out which sites may be contributing more to influence the model performance.

- Agreed

---

## Author Response (AR4)

Response 19 november 2024.

Final reference in the introduction is still incorrectly formatted.
        - We changed it.

Caption for Fig 7 is very brief
        - We elaborated.

Delete or add reference to replace (?) in section 4.5
        - There is no ? anymore

For figure 2(b), check if your figures containing maps/aerial images and photos require a copyright statement/image credit and add it to the figures (or captions). If these figures were entirely created by you, there is no need to add a copyright statement or credit. In that case it is important that you confirm this explicitly by email.
        - We have gathered the first 6 pictures ourselves. The rest are credited to the correct sources.

Also, in future when compiling the tracked changes version please make sure that you have your 'old' and 'new' versions the right way round - the new changes were coming up as deletions in the uploaded file.
        - I am sorry for this, but for some reason latexdiff throws error when doing it the other way around.